# Distinct neuronal populations contribute to trace conditioning and extinction learning in the hippocampal CA1

Rebecca A Mount[1†], Sudiksha Sridhar[1†], Kyle R Hansen[1†], Ali I Mohammed[1], Moona Abdulkerim[1], Robb Kessel[1], Bobak Nazer[2], Howard J Gritton[1]*, Xue Han[1]*

[1]Department of Biomedical Engineering, Boston University, Boston, United States; [2]Department of Electrical and Computer Engineering, Boston University, Boston, United States

**Abstract** Trace conditioning and extinction learning depend on the hippocampus, but it remains unclear how neural activity in the hippocampus is modulated during these two different behavioral processes. To explore this question, we performed calcium imaging from a large number of individual CA1 neurons during both trace eye-blink conditioning and subsequent extinction learning in mice. Our findings reveal that distinct populations of CA1 cells contribute to trace conditioned learning versus extinction learning, as learning emerges. Furthermore, we examined network connectivity by calculating co-activity between CA1 neuron pairs and found that CA1 network connectivity patterns also differ between conditioning and extinction, even though the overall connectivity density remains constant. Together, our results demonstrate that distinct populations of hippocampal CA1 neurons, forming different sub-networks with unique connectivity patterns, encode different aspects of learning.

*For correspondence:
hgritton@bu.edu (HJG);
xuehan@bu.edu (XH)

†These authors contributed equally to this work

Competing interests: The authors declare that no competing interests exist.

## Introduction

The hippocampus is critical for learning and memory in animals and humans. Early surgical lesions of the hippocampus in human patients, designed to alleviate intractable epilepsy, resulted in severe memory loss and an inability to form new declarative or episodic memories (*Scoville and Milner, 1957*; *Scoville, 1954*). Hippocampal atrophy is also associated with diseases related to memory loss and cognitive decline, including dementia and Alzheimer's disease (*Fein et al., 2000*; *Apostolova et al., 2010*; *Chow et al., 2012*; *Henneman et al., 2009*; *Camicioli et al., 2003*). Many mechanistic studies have highlighted the importance of the hippocampus for spatial, contextual, and associative learning in a variety of animal models (*Wirth et al., 2009*; *Jarrard, 1993*).

Various experimental paradigms have been devised to probe hippocampal-dependent forms of learning and memory. One such well-established paradigm is trace eye-blink conditioning, which requires an intact hippocampus (*McEchron et al., 1998*; *Moyer et al., 2015*; *Tseng et al., 2004*). In this experimental design, subjects are presented with a conditioned stimulus (CS), such as a tone or light, which reliably predicts an unconditioned stimulus (US), such as a puff of air or electrical shock delivered to the subject's eyelid. In trace conditioning, the CS and US are separated temporally by a quiescent memory trace interval. Over time, subjects learn to associate the CS with the US, generating a behavioral conditioned response to the CS (*Gruart and Delgado-García, 2007*; *Li et al., 2008*; *McLaughlin et al., 2002*; *Kishimoto et al., 2001*; *Kronforst-Collins and Disterhoft, 1998*; *Takehara-Nishiuchi and McNaughton, 2008*). Trace conditioning acquisition is thought to depend on signaling at both nicotinic and muscarinic acetylcholine receptors (*Brown et al., 2010*; *Disterhoft et al., 1999*; *Fontán-Lozano et al., 2005*; *Raybuck and Gould, 2010*; *Woodruff-*

*Pak, 2003*; *Woodruff-Pak et al., 2007*; *Flesher et al., 2011*; *Woodruff-Pak et al., 1994*) and involves NMDA receptor-dependent plasticity (*Sakamoto et al., 2005*).

The hippocampus is also required for context-dependent extinction learning (*Moyer et al., 2015*). Traditionally, extinction learning is considered new learning that overrides a previously learned relationship. In the example of trace conditioning, the subject learns that the previously established CS is no longer predictive of a subsequent US. Extinction learning after trace conditioning can be tested by presenting the CS without the associated US and monitoring the strength or presence of a conditioned response. As new learning occurs, subjects suppress their learned response to the previously predictive CS. Extinction learning has also been shown to be NMDA receptor-dependent (*Dillon et al., 2008*) and requires the involvement of hippocampal inhibitory neurons (*Lissek et al., 2017*) and adult neurogenesis (*Catlow et al., 2013*).

While the hippocampus is known to be important in both conditioning and extinction learning, it is unclear how individual hippocampal neurons participate in these two types of learning and how neurons interact as learning emerges. Immediate early gene and synaptic tagging experiments revealed that in both the CA1 and dentate gyrus distinct populations of neurons were activated in fear conditioning and context-dependent fear extinction (*Tronson et al., 2009*; *Lacagnina et al., 2019*). While these studies suggest that distinct learning processes are encoded by different subsets of the neuron population, these experiments relied on quantification at later time points, after learning occurs. Such findings, while informative, cannot differentiate neuronal population changes that occur during learning from changes that occur as a result of plasticity in the minutes to hours after learning. To address this question, Zhang et al. recently identified a population of CA1 neurons that emerges to encode contextual fear conditioning using calcium imaging, but they did not examine whether different neuron populations were actively recruited during extinction training (*Zhang et al., 2019*). In order to better understand the mechanisms of evolving interactions between these two types of learning, we performed calcium imaging to measure the ongoing neuronal activity of individual CA1 neurons in mice during both trace eye-blink conditioning and subsequent extinction learning.

Calcium imaging allows us to measure several hundreds of neurons simultaneously with single-cell resolution across many trials, and in the same brain area over multiple days of learning (*Mohammed et al., 2016*; *Hansen et al., 2018*). In our experimental design, once robust conditioning was achieved, mice underwent a final conditioning session immediately followed by extinction training, enabling us to track the same neuron population during both conditions to reveal how extinction learning alters single cell and population encoding in the hippocampus. Hippocampal-dependent trace conditioning is well-suited to calcium imaging because generally both learning and the associated CA1 neuronal responses evolve gradually, unlike fear conditioning, in which learning can occur as rapidly as a single trial.

Our results indicate that different individual CA1 neurons showed CS-related responses during either trace conditioning or extinction learning, suggesting two functionally distinct sub-populations of cells within hippocampal CA1. To further understand how the CA1 network reflects learning as it is occurring, we analyzed co-activity between CA1 neuron pairs on a trial-by-trial basis. We found that distinct pairs of neurons are activated during trace conditioning versus extinction learning, highlighting differential network activity during these two learning processes.

## Results

### Conditioned responding increases across trace conditioning sessions in a classical eye-blink task and decreases during extinction learning

Trace conditioning experiments were performed in head-fixed mice (n = 9 mice) that were positioned under a custom wide-field microscope equipped with a scientific (sCMOS) camera, as previously described (*Mohammed et al., 2016*; *Figure 1A*). Calcium activity in CA1 neurons was monitored via GCaMP6f fluorescence, which allows recording from hundreds of neurons simultaneously (*Mohammed et al., 2016*; *Chen et al., 2013*; *Gritton et al., 2019*). Prior to imaging, mice were injected with AAV-Synapsin-GCaMP6f and implanted with a custom window that allowed optical access to dorsal CA1 (*Figure 1C*). 4–6 weeks after surgery, mice were habituated and trained on a classical trace eye-blink conditioning paradigm followed by an extinction training session

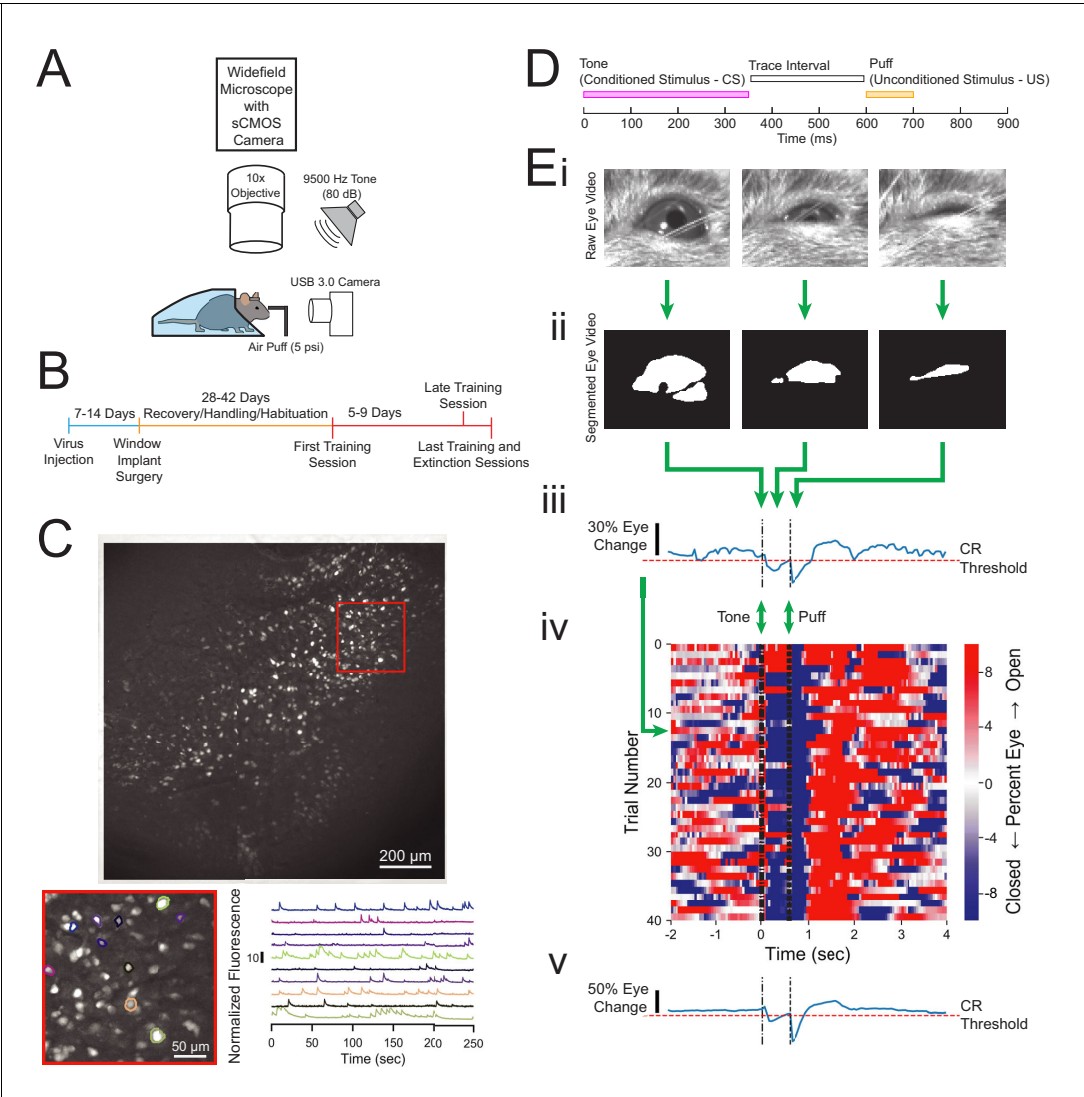

**Figure 1.** Experimental design and quantification of animal behavior. (**A**) Imaging and behavioral setup. The imaging setup consisted of a microscope with a sCMOS camera, standard wide-field fluorescence optics, and a 10× long working distance objective to image a head-fixed mouse. For the behavioral paradigm, a speaker was positioned near the mouse and a cannula for directing an air puff was placed in front of one eye. Eye responses were monitored using a USB 3.0 Camera. (**B**) Experimental timeline. Each animal was injected with AAV-Syn-GCaMP6f and allowed 1–2 weeks for virus expression before surgical window implantation above CA1. The first training session was 4–6 weeks after surgery, and animals were trained and imaged for 5–9 days. (**C**) Full field of view and selected extracted traces. Maximum-minus-minimum projection image for one motion corrected video to show example field of view of several hundred cells. Inset: several selected cells and their corresponding normalized fluorescence trace recordings. (**D**) Within trial design. Trials consisted of a 350 ms tone as the conditioned stimulus (CS), followed by a 250 ms trace interval with no sound, followed by a 100 ms puff of air to the eye as the unconditioned stimulus (US). (**E**) Video eye monitoring and segmentation. (**Ei**) Raw eye frames aligned to the CS, trace interval, and US windows shown above. (**Eii**) Eye frames after segmentation. (**Eiii**) Extracted eye trace and conditioned response (CR) threshold for one trial. (**Eiv**) Eye trace for all 40 trials of a first training session from one example mouse. Red indicates eye opening, while blue indicates eye closure. (**Ev**) Extracted eye trace averaged over all trials shown in Eiv.

The online version of this article includes the following figure supplement(s) for figure 1:

**Figure supplement 1.** GCaMP6 fluorescence trace normalization and rationale.

(*Figure 1B*). The conditioning paradigm consisted of a 9500 Hz, 350 ms tone as a CS, followed by a 250 ms trace interval, which was then followed by a 100 ms gentle puff of air to one eye that served as the US (*Figure 1D*). Eye behavior was monitored with a USB 3.0 Camera (*Figure 1A, Ei*). Animals were trained for 60–80 CS-US trials over 5–9 days until they reached conditioned response criterion (anticipatory eyelid movement on 65% of trials, described below). After reliable conditioned

response to CS presentations was established, on the final day of imaging, animals were given a block of 20–40 CS-US trials, followed by a block of 40 CS-only trials, where the CS was not followed by the US (*Figure 1B*). With this behavioral design, we could perform calcium imaging of the same neurons during both tasks, allowing us to track how the activity of each neuron changes during extinction acquisition. To compare across different learning conditions, the final 20 conditioning trials were analyzed in the first imaging day (first training session) and the last imaging day (last training session). Because extinction learning emerges over time, we analyzed both the first 20 and final 20 CS-only extinction trials (extinction session trials 1–20 and trials 21–40, respectively) (*Figure 1B*).

Behavioral responses were quantified by first segmenting the eye videos to calculate eye area in each frame (*Figure 1Ei, ii*). An eye closure was identified if the eye area dropped below the threshold determined based on the distribution of eye area throughout the entire session. Eye closure between the tone onset, and the puff onset (tone-puff window), was classified as a conditioned response (*Figure 1Ei–iv*). Using this method, we were able to track the strength of the conditioned response to the CS, as well as the strong persistent eye closure in response to the aversive US on each trial (*Figure 1Eiv*). This method also allows for consistent calculation of conditioned responses within each training session (*Figure 1Eiv, v*) and across days (*Figure 2A*).

We first estimated basal spontaneous eye closure occurrence by examining the 'non-stimulus periods' of each trial, defined as the time periods that are more than 2 s before a tone onset or more than 8 s after a tone onset. Spontaneous eye closures occurred in about 20% of the 600 ms windows analyzed during the non-stimulus periods and remained stable across sessions (*Figure 2— figure supplement 1*). For all animals, conditioned eye closure occurrence during the first session was significantly higher than the spontaneous eye closure occurrence (p=0.0003, n = 9 mice, Wilcoxon rank-sum test, *Figure 2—figure supplement 1A*). While most animals (n = 7 mice) showed gradual learning acquisition over the first training session with an eye closure occurrence of 30–70% during the tone-puff window, two animals exhibited rapid learning and correctly responded to over 75% of first session trials. After reaching conditioned response criterion over several days of training, all animals maintained a conditioned response occurrence of 55–90% during the last training session, again significantly higher than the spontaneous eye closure occurrence (p=0.0003, n = 9 mice, Wilcoxon rank-sum test, *Figure 2—figure supplement 1B*).

In order to examine how CA1 neural activity changes during learning, we verified conditioned learning by comparing behavioral responses between the first and last days of training. To most accurately capture the effects of learning that occurs after many days of training, we excluded the two rapid learners that achieved over 75% conditioned responding during the first session from analysis involving comparisons between the first and last training sessions. The remaining seven animals showed significantly more conditioned responding during the last training session (68.9 ± 11.9%) compared to the first training session (45.0 ± 16.8%, p=0.025, n = 7 mice, Wilcoxon rank-sum test, *Figure 2A, Bi*). Next, we analyzed performance during the extinction session and found that conditioned responding trended downward throughout the extinction session (*Figure 2Bii*). By the end of the extinction session, most animals exhibited significantly reduced conditioned responding compared to the last training session. Across all animals, the conditioned response dropped to 47.8 ± 21.1% during trials 21–40 of the extinction session, significantly lower than the conditioned response rate during the last training session (p=0.034, n = 9 mice, Wilcoxon rank-sum test, *Figure 2A, Bii*).

## Calcium dynamics in CA1 neurons track behavioral responses during trace conditioning

To evaluate how CA1 neural activity is modulated between the first and last sessions of conditioning training, we imaged a large number of individual CA1 neurons during trace conditioning (3241 and 2332 total neurons recorded in the first and last/extinction sessions respectively, *Supplementary file 1*). When extinction was introduced, we imaged the same neurons during trace conditioning and extinction learning, enabling us to investigate whether conditioning and extinction recruit unique cell populations or repurpose the same population. In order to assess the activity of individual cells, calcium fluorescence videos were first motion corrected, and then a projection image was generated across each video for cell segmentation using a semi-automated software. Fluorescence traces for each cell were extracted by averaging the fluorescence intensity across all pixels within the cell and normalized for each neuron across each imaging session. To visualize the population response, neuronal responses for each cell were averaged across all trials, and the entire population was sorted by

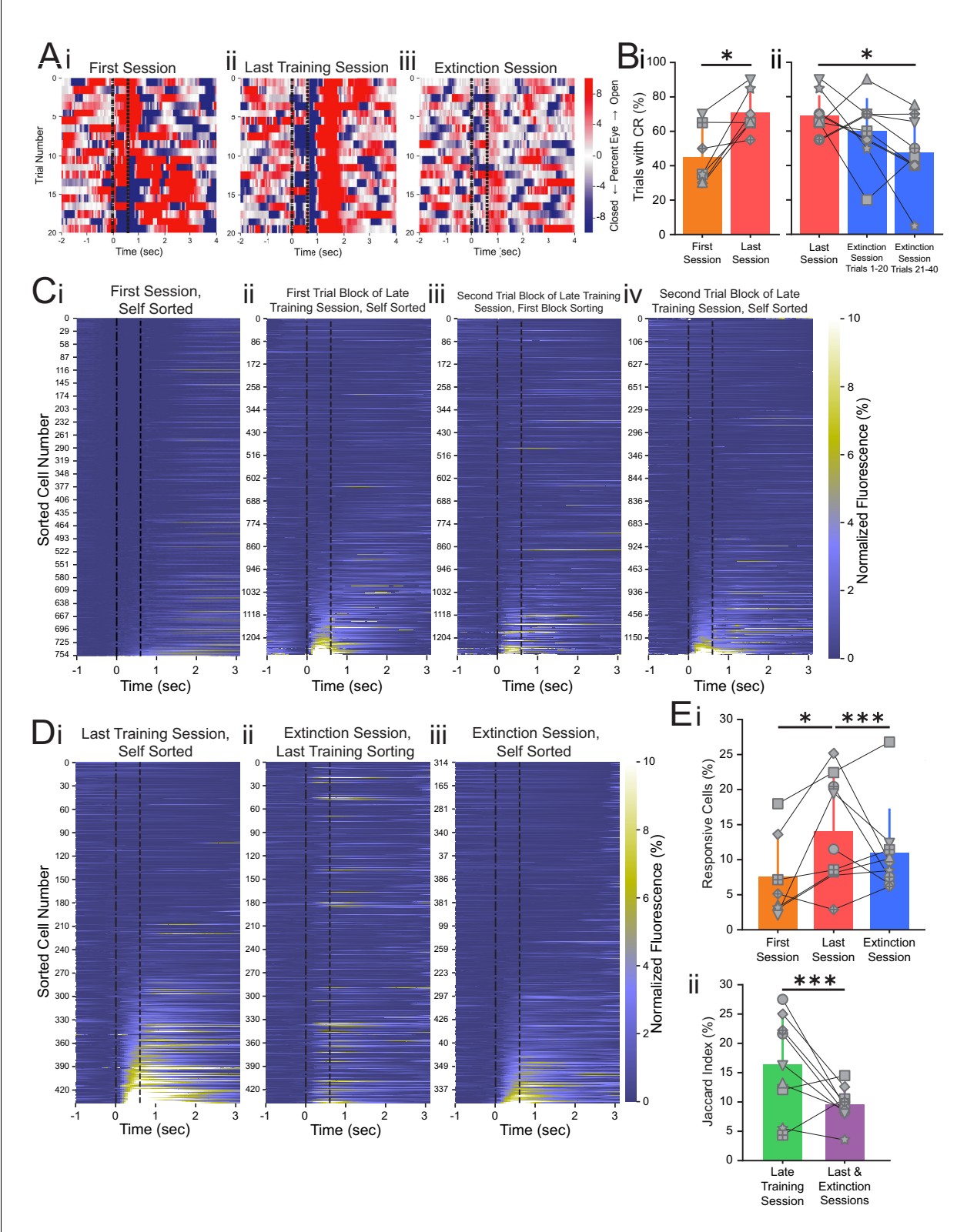

**Figure 2.** Conditioned responses (CRs) and neuronal calcium responses increase during conditioning and decrease during extinction. (A) Extracted eye traces across days. Red indicates an increase in eye area, while blue indicates a reduction in eye area. (Ai) Eye trace for 10 tone-only trials and the first 10 trials of the first training session from the same example mouse in *Figure 1E*. (Aii) Eye trace for the last training session for the same example mouse. (Aiii) Eye trace for the extinction session for the same example mouse. (B) Quantification of CR. (i) Percentage of trials with a CR during the first

*Figure 2 continued on next page*

*Figure 2 continued*

session (orange) and last training session (red), *p=0.025, n = 7 mice for first vs. last session. (ii) Percentage of trials with a CR during the last session (red), trials 1–20 of the extinction session (blue), and trials 21–40 of the extinction session (blue). p=0.37, n = 9 mice for last session vs. trials 1–20 of extinction session, *p=0.034, n = 9 mice for last session vs. trials 21–40 of extinction session, Wilcoxon rank-sum test. (C) Trial-averaged calcium recordings for the first session and late training session. (Ci) Trial-averaged recordings sorted by average fluorescence between the tone and the puff for the first training session from an example mouse. (Cii) Trial-averaged recordings (plotted as in Ci) for the first trial block of the late training session for the same mouse, sorted by average fluorescence between the tone and the puff for the first trial block of the late training session. (Ciii) Trial-averaged recordings (plotted as in Ci) for the second trial block of the late training session, with cell sorting maintained from the first trial block of the session to identify spatially matched cells. (Civ) The same data as shown in Ciii, but resorted according to the fluorescence between the tone and the puff for the second trial block of the late training session. (D) Trial-averaged calcium recordings for the last training session and extinction session. (Di) Trial-averaged recordings (plotted as in C) for the last training session from the same mouse, sorted by average fluorescence between the tone and the puff for the last training session. (Dii) Trial-averaged recordings (plotted as in Di) of the extinction session, with cell sorting maintained from the last training session to identify spatially matched cells. (Diii) The same data as shown in Dii, but resorted according to the fluorescence between the tone and the puff for the extinction session. (E) Quantification of responsive cell properties. (Ei) Percentage of total cells that are identified as tone-responsive for the first session (orange), last training session (red), and extinction session (blue), **p=0.017, n = 7 mice for first vs. last training session, p=0.92, n = 7 mice for first vs. extinction session, ***p=0.0005, n = 9 mice for last training vs. extinction, Fisher's exact test. (Eii) Percentage of cells that are present within both responsive cell populations of the first half and last half of the late training session (green) and both responsive cell populations of the last training session and extinction session (purple), ***p=1.88e-6, n = 9 mice, Fisher's exact test. For all bar plots, error bars are ± s.d.

The online version of this article includes the following figure supplement(s) for figure 2:

**Figure supplement 1.** Conditioned vs. spontaneous eye closure responses.
**Figure supplement 2.** GCaMP6f fluorescence and calcium events in an example recording session.
**Figure supplement 3.** Calcium responses from all neurons recorded in all animals.
**Figure supplement 4.** Calcium response latency from all neurons recorded in all animals.
**Figure supplement 5.** GCaMP6 fluorescence averaged every 10 trials throughout an example recording session.

average response intensity during the time period between the tone onset and puff onset (tone-puff window, *Figure 2C*).

To quantify changes in neural responses, individual calcium events were identified in each normalized fluorescence trace (*Figure 2—figure supplement 2A, B*). Consistent with previous observations of sparse calcium events (*Gritton et al., 2019*), we found that the basal calcium event rate was $1.7 \pm 0.4$ events/min ($0.03 \pm 0.006$ events/s) during the non-stimulus periods (n = 297,336 events, across n = 9 animals from n = 27 sessions). We then binarized calcium traces by assigning ones to the rising phases of all calcium events and zeros to the rest of the trace. To estimate the basal spontaneous calcium event amplitude distribution for each trace, we randomly shuffled the non-stimulus periods of the binarized calcium trace 1000 times for each cell in each imaging session. In each shuffle, we randomly selected 20 non-overlapping segments from the non-stimulus periods of each cell in the imaging session, calculated the mean calcium event amplitude from these 20 windows, and plotted this mean calcium event amplitude to form the basal distribution for that cell. To determine if a cell is responsive to the tone, we then calculated the mean calcium event amplitude within the 1 s windows following the 20 tone onsets. If a cell's mean tone-relevant calcium event amplitude was greater than the 95th percentile of the basal distribution for that cell, that cell was identified as a 'responsive cell'.

After multiple days of conditioning, 14.7% of neurons met the criteria as responsive to the tone during the last training session (286 responsive neurons of total 1946 recorded neurons in seven mice, *Supplementary file 2*), significantly greater than the responsive population during the first training session (12.2%, 288 responsive neurons of total 2367 recorded neurons from seven mice, p=0.017, Fisher's exact test, *Figure 2Ci*, Di, Ei, *Supplementary file 2*, *Figure 2—figure supplement 3A*; *Figure 2—figure supplement 3B*). In addition to the increase in the fraction of responsive cells after conditioning training, calcium response latency was significantly shorter during the last training session ($337.6 \pm 202.5$ ms) as compared to the first training session ($436.9 \pm 202.9$ ms, p=0.001, one-way ANOVA post-hoc Tukey test, n = 795 neurons, *Figure 2—figure supplement 4*). Shorter calcium response latency indicates that with repeated training CA1 neurons shift their responsivity closer to the CS tone presentations, further confirming more robust CA1 responses after trace conditioning. Consistent with previous studies (*Zhang et al., 2019*), our findings demonstrate that CA1 neurons begin to encode the CS on the first day of training, and they more reliably encode the CS over several days of continued conditioning training.

## Extinction learning rapidly recruits new CA1 neurons

As described above, the last imaging day included both the last conditioning session and the extinction session, thus allowing us to track the same cells throughout both sessions. Behavioral analysis revealed that conditioned responding was significantly reduced during the extinction session as compared to response rate during the last training session (*Figure 2B*). To visualize the responses of individual cells to the CS during conditioning trials and extinction trials, we plotted the neuronal responses of the entire population during each session (*Figure 2D*). We discovered that many neurons with strong responses to the tone during conditioning do not respond to the tone during the extinction session (*Figure 2Di, ii, Eii*). When we re-sorted the neuronal responses based on their responses during the extinction session, we found a largely separate population of neurons that were responsive during extinction uniquely (*Figure 2Diii*). To visualize the temporal dynamics of response changes across the population, we plotted the average responses of individual CA1 neurons every 10 trials (*Figure 2—figure supplement 5Ai, ii*). Over the 20 trials of the last training session, the responsive CA1 population remained relatively constant between the two blocks of 10 trials. However, CA1 neuronal responses changed with extinction training, and a separate responsive CA1 population emerged within the first 20 extinction trials (*Figure 2—figure supplement 5Aiii–vi, B*). To capture these rapid population changes, we used the first 20 CS-only extinction trials as the 'extinction session' for all analyses of neuronal dynamics.

We found that 11.4% of neurons were responsive to the tone during the first 20 trials of the extinction session (266 responsive neurons from total 2332 recorded cells in nine mice, *Supplementary file 3*), significantly less than the 14.9% responsive neurons during the last training session and similar to the 12.2% responsive neurons during the first session (p=0.0005, n = 9 mice for last session vs. extinction session and p=0.92, n = 7 mice for first session vs. extinction session, Fisher's exact test, *Figure 2Ei*, *Figure 2—figure supplement 3*, *Supplementary file 3*). The reduction in responsive neurons during the extinction session is likely a reflection of the network remodeling that is occurring during extinction learning, which would not have been present in the late training session as conditioned learning was well-established during that session. However, the existence of this responsive population during the extinction session demonstrates that neurons in CA1 show new encoding of the CS during extinction learning, after it is no longer paired with the US. To compare the identities of neurons during trace conditioning and extinction learning, we classified conditioned (CO) cells as those that responded to the CS during the last training session and extinction (EX) cells as those that responded to the CS during extinction. As behavioral performance and neural activities could vary throughout each training session, we estimated the chance of detecting discrete populations over the course of an imaging session by analyzing the imaging session prior to the last training session (late training session, *Figure 1B*). We analyzed two blocks of 20 CS-US trials (trials 1–20 and trials 21–40) of the late training session. To quantify the proportion of common responsive cells between sessions or trial blocks, we calculated the Jaccard index, defined as the number of cells responsive in both sessions divided by the total number of cells responsive in either session. The Jaccard index for the common responsive cells between the two trial blocks of the late training session was 20.5%, significantly greater than the Jaccard index observed between the last training and extinction sessions (10.7%, p=1.88e-6, n = 9 mice, Fisher's exact test, *Figure 2Eii*, *Supplementary file 5*). Together, these results suggest that during extinction learning, in less than 20 trials, a largely unique population of neurons is recruited to encode tone presentations.

## Temporally and spatially distributed populations of neurons encode either trace conditioning or extinction learning

Because calcium events are sparse, we next considered the heterogeneity of individual cell activation during conditioning and extinction learning. We analyzed the number of trials that CO and EX cells exhibited calcium event onsets in the 1 s following tone onset. While some cells showed calcium responses to the CS on a large number of conditioning or extinction trials (example CO cells: *Figure 3A*; example EX cells: *Figure 3B*), most CO and EX cells respond to a small number of trials. CO cells exhibited calcium events on 13.3 ± 2.3% of trials during the last training session, and EX cells exhibited calcium events on 7.0 ± 1.0% of trials during the extinction session. Because most neurons individually contributed to encoding only a small fraction of trials, trial encoding may thus depend on the contributions of a large population of sparsely active neurons.

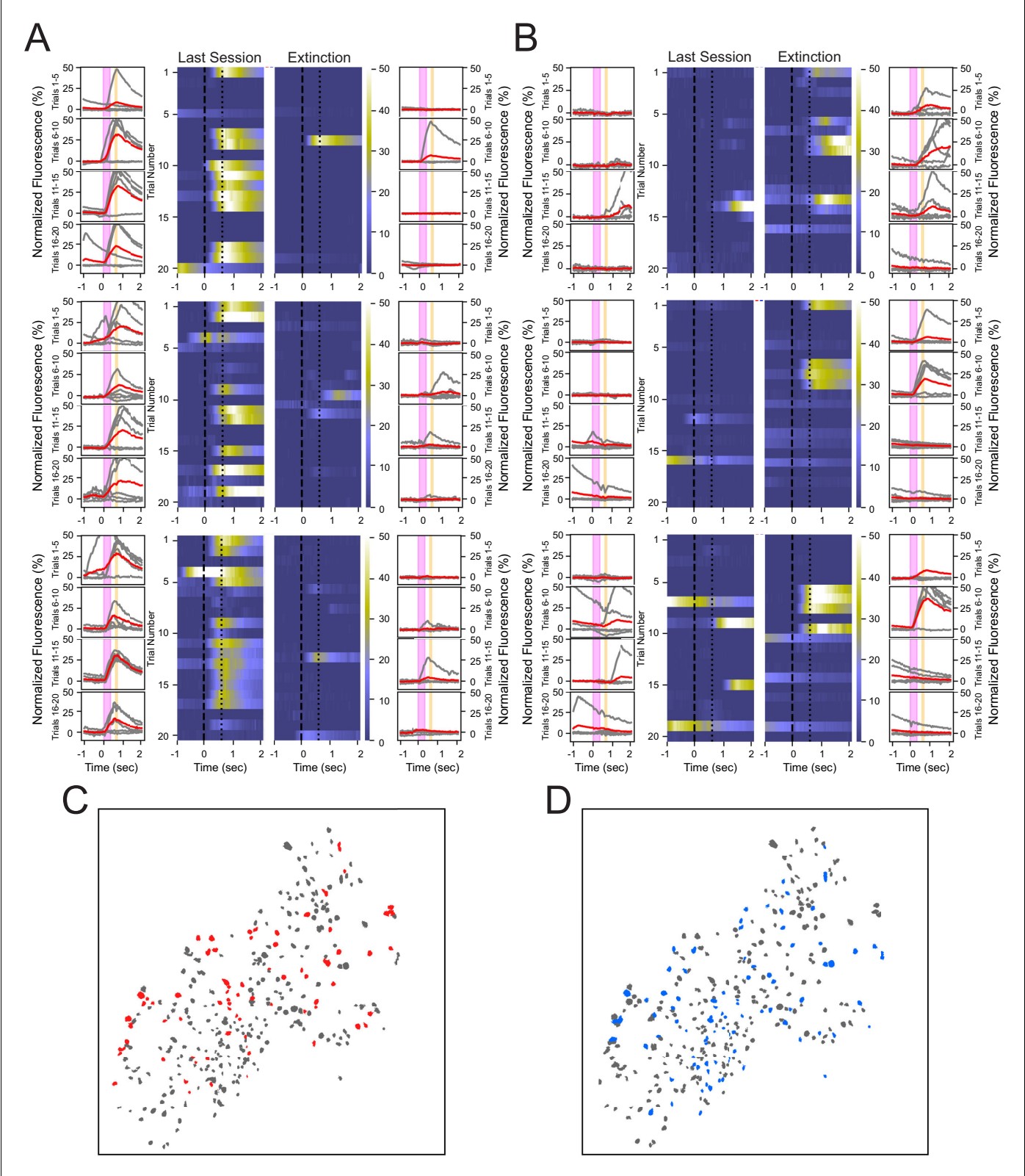

**Figure 3.** Heterogeneous neuronal responses to the tone during conditioning and extinction learning. (**A**) Responses across trials for three neurons that show reliable responding during the last training session, but not during the extinction session, termed CO cells. Outer columns are individual trials shown in gray, and the average of five trials shown in red. The pink box corresponds to the tone, and the orange box corresponds to the puff. Heat maps in the center show each trial for a 3 s time window surrounding the tone and puff presentations. (**B**) Three neurons that exhibit reliable

*Figure 3 continued on next page*

*Figure 3 continued*

responding during the extinction session, but not during the last training session, termed EX cells. (**C, D**) Spatial maps of all neurons from a representative animal during last conditioning training session (**C**) and extinction session (**D**), with CO cells in red, EX cells in blue, and all other cells in gray.

Comparing the spatial distributions of cells indicates that both CO and EX cells are not significantly clustered near one another and are heterogeneously distributed throughout the hippocampus (*Figure 3C, D*). We found that 3.8 ± 2.9% of cells within a 100 µm radius of CO cells were other CO cells, which was not significantly different from that expected by random shuffling of cell identity (shuffled = 2.8%, N = 1000, p=0.17, two-tailed bootstrap, α = 0.05). Similarly, 2.3 ± 2.6% of cells within a 100 µm radius of EX cells were other EX cells, which was also not significantly different from that expected by random shuffling (shuffled = 1.8%, N = 1000, p=0.24, two-tailed bootstrap, α = 0.05). These analyses reveal that individual CO cells and EX cells responded on a sparse subset of trials, and that CO and EX cells were heterogeneously distributed within CA1 with little spatial organization.

## Co-occurrence analysis reveals differential connectivity between subpopulations of neurons during trace conditioning versus extinction learning

Because CO and EX cells responded on only a sparse subset of trials, but behavioral performance was robust across sessions, we hypothesized that population responses may be critical to the role of the CA1 network in learning and memory. While pairwise correlation can give reliable measures of network activation over many trials, or over longer periods of time, the limited number of imaging data points during the short (600 ms, 12 frames) tone-puff window of this study made pairwise correlation unsuitable for tracking neuronal calcium responses. Therefore, we quantified network responses based on co-activity of cell pairs on a trial-by-trial basis and summarized co-activity across all neuron pairs in a 'co-occurrence matrix.' For each trial, if a neuron exhibited a calcium event onset in the 1 s following tone onset, that neuron was assigned a 1 (*Figure 4Ai*). All other neurons (those without a calcium event onset following tone onset on that trial) were assigned a 0. Taking the outer product of this response vector yielded a co-occurrence matrix of all cell interactions in the population for a single trial (*Figure 4Aii*). These single-trial co-occurrence matrices were then combined over trials and clustered using spectral biclustering to visualize neurons that were co-modulated on those trials (*Kluger et al., 2003*; *Figure 4Aii*). Clusters of co-active neuron pairs in the last training session appeared largely not co-active during the extinction session (*Figure 4Bi, ii*). However, reclustering the extinction session matrix revealed new clusters of co-active neuron pairs on extinction trials (*Figure 4Biii*), consistent with our finding of a new population of cells encoding the tone during extinction learning.

To quantify network connectivity, we anatomically mapped co-activity as edges between cells that were responsive during the last training session (CO cells) or the extinction session (EX cells, *Figure 4Ci*). We found no significant difference in the number of edges present in the last training session versus extinction session (60.0 ± 19.7% vs. 46.4 ± 19.6% of the total edges from the last training and extinction sessions combined, t = 0.98, p=0.36, n = 9 mice, two-tailed paired t-test, *Figure 4Di*). Additionally, the connectivity density (the observed number of edges divided by the total number of possible edges formed between neurons) and degree (two times the average number of edges per neuron) of the two maps were not different from one another (t = 1.78, p=0.11 and t = 1.82, p=0.11 for density and degree, respectively, n = 9 mice, two-tailed paired t-test, *Figure 4—figure supplement 1*).

However, when we overlaid the last session and extinction session maps, we noticed that edges during the last training session appeared largely distinct from edges present during the extinction session (*Figure 4C, Di*). To estimate whether the difference in edge identity between the last session and extinction session is above the chance of detecting discrete networks throughout a training day, we performed the same co-occurrence network analysis on two separate trial blocks (trials 1–20 and trials 21–40) from the late training session. As expected, co-activity amongst many neuron pairs during the first trial block of the late training session was maintained during the second trial block of the session (*Figure 4—figure supplement 2A*). To quantify changes in network edge identity, we

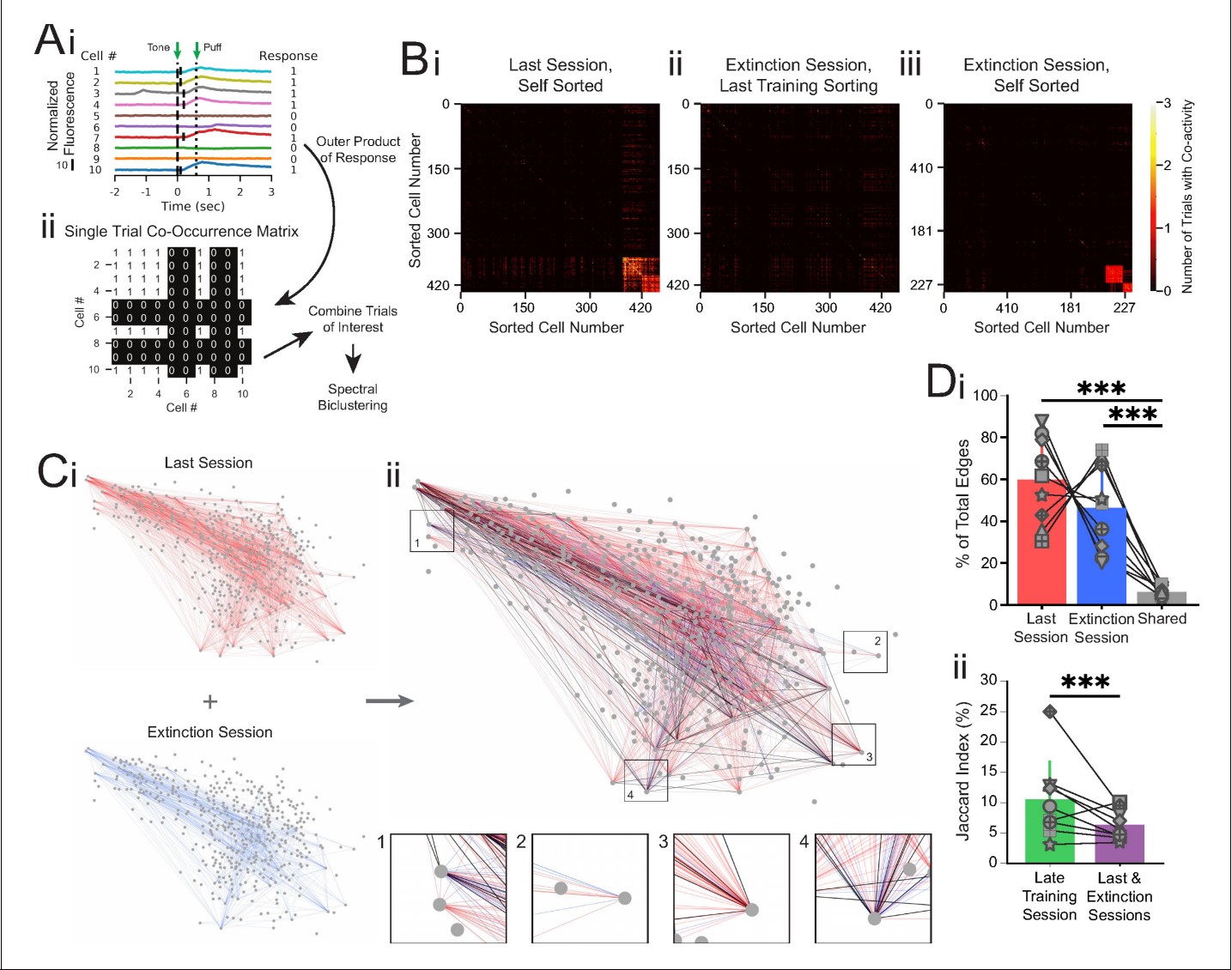

**Figure 4.** Co-occurrence network analysis during last training session vs. extinction session. (**A**) Schematic of method for constructing single-trial co-occurrence matrices. (**Ai**) A sub-population of cells for one trial that highlights how the response pattern was determined. If a cell exhibited a calcium event (denoted by vertical black line at event onset) during the 1 s window following tone onset, it was assigned a 1. (**Aii**) The outer product was taken of the vector of responses across the population with itself to generate a single-trial co-occurrence matrix. This is a binary matrix where if the ith and jth cells both exhibit a calcium event during the 1 s window following tone onset there is a 1, but a 0 otherwise. These individual trials can be combined as specific trials of interest and clustered with spectral biclustering to identify neurons with the highest degree of co-activity across those trials. (**B**) Representative co-occurrence matrices showing clusters of co-active neuron pairs in the last training session (**Bi**) and the extinction session, with sorting maintained from the last training session matrix (**Bii**), and re-clustered results based on the matrix during extinction session (**Biii**). (**C**) Connectivity maps created from co-occurrence matrices for the last training session (**Ci**, top), extinction session (**Ci**, bottom), and overlay (**Cii**). Edges from the last training session are shown in red, edges from the extinction session are shown in blue, and edges present during both sessions are shown in black. Insets: zoom-ins of four nodes. (**Di**) Quantification network edges present during the last training session (red), extinction session (blue), or both (gray). t = 0.98, p=0.36 for last vs. extinction sessions, ***t = 7.74, p=5.5e-5 for last session vs. shared and t = 5.73, p=0.0004 for extinction session vs. shared, n = 9 mice, two-tailed paired t-test. (**Dii**) Percentage of edges that are present in both the first half and last half of the late training session networks (green) vs. the last training session and extinction session networks (purple), ***p=2.48e-8, n = 9 mice, Fisher's exact test. For all bar plots, error bars are ± s.d.

The online version of this article includes the following figure supplement(s) for figure 4:

**Figure supplement 1.** Network degree and density.
**Figure supplement 2.** Networks of two separate trial blocks during the late training session.

calculated the Jaccard index as the common edges between the two sessions divided by the total edges in both sessions. We found a Jaccard index of 8.2% common edges between the network maps for the last training and extinction sessions, significantly smaller than the 10.4% common edges between the networks for the two trial blocks of the late training session (600 shared edges of 7306 total edges in last training/extinction sessions vs. 2159 shared edges of 20676 total edges in late training session, p=2.48e-08, n = 9 mice, Fisher's exact test, *Figure 4C, Dii*, *Figure 4—figure supplement 2B*, *Supplementary file 6*). As individual responsive cells respond on a small number of trials, the common edge Jaccard index between co-active neuron pairs is naturally lower than the common cell Jaccard index described in *Figure 2*. However, the significant decrease in common edge Jaccard index for the last training and extinction sessions in comparison to that of the two trial blocks of the late training session is consistent with the observation of distinct responsive cells during extinction learning. Furthermore, the distinct activation of different cell pairs suggests unique network patterns of activity during each learning condition, while the overall involvement of the CA1 network (connectivity density) remained constant during the conditioning and extinction training sessions.

## Discussion

In this study, we provide the first detailed, real-time evidence that largely distinct populations of neurons within the hippocampal CA1 region respond to a trace conditioned stimulus during either conditioned learning or extinction learning. Previous studies have looked at calcium activity associated with trace conditioning acquisition, but have not investigated whether a separate population of neurons contributes to extinction learning (*Zhang et al., 2019*; *Mohammed et al., 2016*; *Modi et al., 2014*). It has been previously reported that two functionally distinct neuron populations are activated by fear conditioning and extinction in the amygdala (*Herry et al., 2008*). Additionally, a subsequent study looking at the CA1 region of the hippocampus revealed distinct changes in gene phosphorylation states in largely non-overlapping neural populations in either fear conditioning or extinction (*Tronson et al., 2009*). However, in this study, cFos and pERK immuno-activities were used as markers of conditioning and extinction learning and measured hours to days after the respective training. While these results provided the first important insight into the potential for distinct population encoding, the indirect nature of the activity markers and the time course for immuno-quantification does not allow for the distinction between rapid or gradual evolution of conditioning and extinction neuron populations. A more recent synaptic tagging study found populations of cells in the dentate gyrus that are re-activated during either fear conditioning recall or extinction retrieval, presumably representing separate populations (*Lacagnina et al., 2019*). However, these populations were identified using the immediate early gene Arc in separate cohorts of mice, which does not allow for real-time monitoring or direct comparisons between the populations.

In order to better understand the dynamic relationship between conditioning and extinction learning in the hippocampus, and to further investigate whether distinct populations encode these learning events, we used calcium imaging to monitor the activity of individual cells throughout conditioning and extinction learning paradigms. We applied trace conditioning because (1) it lends the advantage of a singular defined stimulus to which neural activity can be easily aligned and measured between the two different paradigms, (2) it allows imaging of the same neurons during both conditioning and extinction in the same imaging session, and (3) learning during trace conditioning evolves over multiple trials, unlike fear-based paradigms where learning often occurs over very few trials. Interestingly, we found that the rate of conditioning was highly animal-dependent, evolving gradually in most animals, but rapidly in a small subset of mice that showed substantial responding to the CS within 40 conditioning trials. Although acquisition of trace eye-blink conditioning can occur over dozens of trials in a single session (*Modi et al., 2014*), most of our animals showed gradual behavioral improvement, reflected in the increase in conditioned response rate after multiple days of training and the increase in the proportion of neurons that responded to the tone from the first to the last training session. Overall, our results support the idea that robust conditioned learning gradually evolves over days as CA1 neurons are recruited to encode the stimulus and reflect previous electrophysiology studies in rabbits and rats where the time course of learning evolves through many CS-US pairings (*Berger et al., 1983*; *Green and Arenos, 2007*).

In our study, extinction learning evolved rapidly, and a new population of neurons that responded to the now-extinguished tone emerged within just 20 extinction trials. Previous work has implicated the prefrontal cortex and septal cholinergic inputs to be critical to the process of extinction, and these pathways may play a pivotal role in the rapidity of extinction neuron emergence (*Acquas et al., 1996*; *Chang et al., 2010*; *Euston et al., 2012*). More work will need to be done on this front to determine whether reducing or silencing these inputs could delay or block the emergence of extinction-selective neuron populations. In addition, it is possible that extinction learning may occur more rapidly because meaningful associations between conditioning cues and outcomes are already established. Studies probing location encoding of familiar places, where a spatial memory schema already exists, suggest that re-encoding of spatial memory occurs more rapidly when a spatial map is already established and the new information is updating that context (*Tse et al., 2007*; *McKenzie et al., 2013*). Accordingly, our observation of rapid emergence of extinction learning could be due to the presence of a contextual representation of the task stimuli prior to extinction training. Since extinction neuron activity can emerge rapidly, the mechanisms of interaction between conditioned neuron and extinction neuron activity may be an important future direction. Such work could benefit the treatment of anxiety-based disorders such as post-traumatic stress disorder, which is characterized by over-generalization of fearful stimuli to neutral contexts and impairments in the development of extinction learning (*Orr et al., 2012*; *Maeng and Milad, 2017*; *Burriss et al., 2007*; *Wessa and Flor, 2007*; *Garfinkel et al., 2014*; *Milad et al., 2009*).

Calcium imaging is a powerful tool to understand how large populations of neurons function at the network level. However, when investigating dynamic or rapid network changes, as in extinction learning, it can be difficult to decode the information present in the population using traditional analysis techniques. For example, pairwise correlation measured over our brief tone-puff window (600 ms, 12 data points) is unreliable on a trial-by-trial basis. Traditional single-trial analytic approaches usually cannot find meaningful correlations with such limited data. Thus, our development of a co-occurrence-based network approach provides a robust way to observe the trial-by-trial evolution of the population responses and a means to assess contributions of certain cells that might be otherwise overlooked, or overstated, in trial-averaged data. Additionally, co-activity allows us to consider functional connectivity maps of entire neuron populations, an intuitive way to visualize and investigate the patterns of neural activation. Overall, co-occurrence matrix analysis is a useful technique for monitoring the evolution of population responses over time from high-dimensional calcium imaging datasets.

Using co-occurrence matrices, we found that CA1 neurons' connectivity patterns change drastically between conditioned trial performance and extinction learning. While some cells may participate in both learning conditions, unique pairs of neurons are differentially activated during the different types of learning, indicating a role of network activation in CA1. However, the network connectivity density and degree remain constant across these different behavioral paradigms, highlighting the constant overall involvement of the CA1 network during both learning tasks. Overall, our results provide important insight into how the hippocampus may represent and encode behaviorally relevant stimuli.

# Materials and methods

## Animal surgery and recovery

All animal procedures were approved by the Boston University Institutional Animal Care and Use Committee. A total of nine female C57BL/6 mice, 8–12 weeks old at the start of the experiments, were used in these studies (Taconic; Hudson, NY). To estimate sample size, power analysis was based on effect size differences found in our previous trace conditioning calcium results recorded in the hippocampus (*Mohammed et al., 2016*). Power analysis was performed using G*Power 3.1.9.6 (http://www.gpower.hhu.de), applying a one-tailed Wilcoxon signed-rank test utilizing α = 0.05 and a power (1-β) value of 0.80, and yielded a sample size of 9. Following arrival from the vendor, mice were group housed and allowed to habituate to the vivarium for 2+ weeks prior to surgery. Animals first underwent viral injection surgery targeting the hippocampus under stereotaxic conditions (AP: −2.0 mm, ML: +1.5 mm, DV: −1.6 mm). Mice were injected with 250 nL of AAV9-Synapsin-GCaMP6f.WPRE.SV40 virus obtained from the University of Pennsylvania Vector Core (titer ~6e12

GC/mL). All injections were made via pulled glass pipettes (diameter: 1.2 mm) pulled to a sharp point and then broken at the tip to a final inner diameter of ~20 μm. Virus was delivered via slow pressure ejection (10–15 psi, 15–20 ms pulses delivered at 0.5 Hz). The pipette was lowered over 3 min and allowed to remain in place for 3 min before infusion began. The rate of the infusion was 100 nL/min. At the conclusion of the infusion, the pipette remained in place for 10 min before slowly being withdrawn over 2–3 min. Upon complete recovery (7+ days after virus injection), mice underwent a second procedure for the implantation of a sterilized custom imaging cannula (OD: 3.17 mm, ID: 2.36 mm, height, 2 mm diameter), fitted with a circular coverslip (size 0; OD: 3 mm) adhered using a UV-curable optical adhesive (Norland Products). To access dorsal CA1, the cortical tissue overlying the hippocampus was carefully aspirated away to expose the corpus callosum. The white matter was then thinned until the underlying tissue could be visualized through the surgical microscope. The window was then placed and centered above the hippocampus. During the same surgery, a custom aluminum headplate was attached to the skull, anterior to the imaging cannula.

## Animal training and trace conditioning behavioral paradigm

Animals were allowed at least 2 weeks to recover from window surgeries, followed by an additional 2–4 weeks of handling and habituation to being head-fixed underneath the microscope (*Figure 1B*). Each animal received at least three habituation sessions prior to the first imaging day. Habituation was performed in the dark with the imaging LED illuminated to the same intensity as it would be for recording sessions.

Following habituation, mice were trained on a trace eye-blink conditioning task similar to what has been described previously (*Mohammed et al., 2016*). Each trial consisted of a 350 ms long 9500 Hz tone (CS) at 78–84 dB followed by a 250 ms trace interval, followed by a 100 ms puff to the eye (US) at 4.2–6 psi (*Figure 1Bi*). The ambient noise level ranged between 55 and 60 dB. Inter-trial intervals for each presentation were pseudo-randomized within a recording session with an inter-trial interval of 35 ± 5 s. Animals were first presented with 20 tone-only trials. Animals were then presented with either 60 tone-puff trials per day for 8 days or 80 tone-puff trials per day for 4 days. The final recording session consisted of 20 or 40 tone-puff conditioning trials, followed by 40 extinction trials, where the puff was removed but the tone remained. For all analyses, we analyzed the final 20 trials of the first imaging day (first training session) and the final 20 CS-US conditioning trials on the last imaging day (last training session). For behavior analysis, we analyzed the first 20 and final 20 CS-only extinction trials on the last imaging day, and for analysis of neuronal dynamics we analyzed the first 20 CS-only trials on the last imaging day (extinction session).

Behavioral stimuli were generated using a custom MATLAB script that delivered TTL pulses for the tone and puff via an I/O interface (USB-6259; National Instruments, Austin, TX). Behavioral TTL pulses and imaging frame timing were digitized and recorded (Digidata 1440A; Axon CNS Molecular Devices, San Jose, CA, or RZ5D Bioamp Processor; Tucker Davis Technologies, Alachua, FL) to align behavioral data and imaging frames. Mouse eye behavior was captured using a Flea3 USB 3.0 camera (FL3-U3-13S2C-CS; Richmond, BC, Canada) and the Point Grey FlyCapture2 software. The mouse's face and eye were illuminated with an infrared lamp positioned approximately 0.05–0.5 m away from the mouse.

## Wide-field imaging

A custom wide-field microscope was used to record neuronal calcium responses during animal learning and behavior as previously described (*Mohammed et al., 2016*). Briefly, the animal was head-fixed below the microscope on an articulating base (SL20 Articulating Base Ball Stage; Thorlabs Inc, Morganville, NJ) via a custom-machined attachment for the headplate. The animal was gently wrapped by an elastic self-adherent wrap to reduce movement during recording. The microscope consisted of a scientific CMOS (sCMOS) camera (ORCA-Flash4.0 LT Digital CMOS camera C11440-42U; Hamamatsu, Boston, MA), standard optics for imaging GCaMP6f, and a ×10 magnification objective (Leica N Plan 10 × 0.25 PH1 or Mitutoyo Plan Apo Long WD Objective 10 × 0.28). Images yielded a field of view 1.343 mm by 1.343 mm (1024 × 1024 pixels) and were acquired at a 20 Hz sampling rate and stored offline for analysis.

## Data analysis

All custom software is available on GitHub (github.com/HanLabBU/Distinct-populations-hippocam-pus) (*Sridhar and Hansen, 2021a*; copy archived at swh:1:rev:960e4b4d92e42697649b9-b9a684ecf9c4cbb79f6 ;*Sridhar and Hansen, 2021b*).

## Eye-blink segmentation and analysis

Each raw eye video was segmented in ImageJ (Fiji [*Schindelin et al., 2012*]) using the MorphoLibJ plugin (*Legland et al., 2016*) to generate a binary mask video. Each frame of this binary video was summed and normalized by the average eye area to generate a trace corresponding to the percentage of eye closure over time. First, image stacks were loaded as grayscale images, Gaussian filtered with a radius of 2, and thresholded to include only the eye range. Videos were converted to binary, holes were filled, and the Particle Analyzer feature was used to exclude all regions on the edges of the videos above the thresholded value. The MorphoLibJ plugin (*Legland et al., 2016*) was used to label connected components with a connectivity of 26. A custom Jython script (StepIntegers.py) was used to determine the connected components that existed across all image frames, which were merged into one connected component. Lastly, to capture any additional smaller connected components, which were commonly created at or around the time of eye closures, another custom Jython script (FindModalValues.py) was used to capture these remaining components, which were then merged into the final connected component. All other connected components not a part of this singular merged component were dropped from the binary mask stack, which was saved for eye-closure trace generation.

Eye traces over time were generated by summing the binary pixels corresponding to the segmented eye in each video frame and dividing by the average eye area across the whole video. A conditioned response was classified by a threshold of 2% eye area deviation below the standard eye area. The threshold was calculated by fitting a line to the central 95th percentile of the full eye trace. This threshold is equivalent to when the residuals deviated by 2% from a uniform distribution fit to the eye trace that was equal to the average eye area. Eye traces for six mice were manually inspected by an independent observer to confirm that eyelid movements ('blinks') identified using the 2% threshold matched blinks selected by visual inspection. Each time the eye trace showed a decrease below this threshold between the tone onset and puff onset, that trial was classified as a conditioned response trial.

As stated previously, the final 20 CS-US trials of the first imaging day were chosen for analysis. These trials were chosen to avoid potential startle response to novel stimuli (especially the puff) in the mice, and because calcium response latency was shorter during these trials as compared to the first 20 CS-US trials of the first imaging day. Latency of calcium response was determined as the average onset time of the first calcium event after tone onset across trials for each responsive cell. Average response latency in the first 20 trials of the first imaging day was 470.3 ± 214.1 ms, and by the last 20 trials of the same imaging day, average latency dropped to 436.9 ± 202.9 ms (*Figure 2—figure supplement 4*), suggesting a reduction in response to the eye puff after the first 20 trials of the first imaging day. Shorter response latency is consistent with the idea that the neuronal response is shifting from being puff-responsive to being tone-responsive.

The mice with the highest behavioral performance (=>75% correct response rate) during the first training session were excluded from all analyses comparing first session to last training session or first session to extinction session (n = 2 mice). Their rapid learning of the task prevents the capture of neuronal changes during learning. All mice were included for all analyses comparing last training session to extinction session and all analyses of the late training session.

## Spontaneous eye closure occurrence calculation

Data was split into trial periods (2 s before tone onset to 8 s after tone onset, 201 imaging frames) and non-stimulus periods (all other frames). For spontaneous eye closure occurrence calculation, a sliding window of 12 consecutive points of the eye trace (600 ms, equivalent to the tone-puff window) during the non-stimulus periods was considered at a time. Each window was analyzed as described above; a decrease below the eye area threshold within the window was classified as a spontaneous eye closure.

## Movement correction

Motion correction of videos was done using *ptmc*, an open-source, parallel Python version (github. com/HanLabBU/ptmc) of an image stabilization process published previously (*Mohammed et al., 2016*; *Hansen, 2017a*). Briefly, each frame was motion corrected by median filtering each image to remove noise, homomorphic filtering the image for edge detection, and comparing the frame with a reference image to determine how many pixels to shift that specific frame. This process was run in parallel by first motion correcting the first multi-page tiff stack (2047 frames) to an average projection image of the noisy, non-corrected video. This corrected video stack was used to generate a new reference image that was sent out in parallel with every frame across the whole video, including the first tiff stack used to generate the reference.

## Neuronal trace extraction

After motion correction, regions of interest (ROIs) corresponding to cells were selected using a semi-automated custom MATLAB software called *SemiSeg* (github.com/HanLabBU/SemiSeg) (*Hansen, 2017b*). A projection image (maximum fluorescence minus minimum fluorescence) was calculated across the whole video stack for selecting ROIs. This static frame was loaded into SemiSeg, and the full boundary of the ROIs was selected by a user. This selected sub-region of the image was automatically thresholded to determine the pixels within that region that correspond to a cell. After all cells were selected from the projection image, pixels from each ROI were averaged together spatially to calculate a temporal trace for each neuron.

## Co-registration of neurons

Due to a technical memory limitation on the recording computer, for some mice there was a temporal gap of about 20 min between the last training session and the extinction session, in which neither the mouse nor microscope were moved (n = 3 mice). These recording sessions necessitated co-registration of cells between the sessions, unlike the continuous recording sessions where the camera continuously recorded between the last training session and extinction session (n = 6 mice). For sessions where ROIs were matched to one another, spatial ROI maps were co-registered using framewise cross-correlation. ROIs were then matched using a greedy method that required the centroid of cells to be within 50 pixels of one another and had to have at least 50% of their pixels overlap, as published previously (*Shen et al., 2018*). Cells that did not meet both of those criteria were removed from the matched dataset for comparison.

## Fluorescence trace normalization

A local background trace was calculated for each neuron by finding the centroid for each ROI and measuring a circle approximately 10 cell widths in radius (100 pixels) and subtracting the area for the ROI from that circle. The pixels in this local background were averaged together spatially to measure a temporal background trace. Corresponding background traces were subtracted from each cell's measured trace to remove local fluctuations from scattering in wide-field imaging. After local background subtraction, each neuron's fluorescence trace was normalized. Due to the length of recording sessions, traditional normalization techniques (such as mean baseline subtraction and standard deviation threshold selection) disproportionally affect cells that are more active. Therefore, the baseline calcium level was calculated for each cell by fitting a normal distribution to the lowest 50 percentile of the data (assuming a Gaussian noise fluctuation surrounding a true baseline calcium level) and using the mean of this distribution as the baseline calcium level (*Figure 1—figure supplement 1A*). This baseline was subtracted from each trace (*Figure 1—figure supplement 1B, C*).

The heat maps in *Figure 2* and *Figure 2—figure supplement 5* were generated by averaging the calcium imaging data across trials in a session for an individual mouse. *Figure 2—figure supplement 4* was generated in a similar manner for all the cells from all mice. The average fluorescence of the 12 data points (600 ms, equivalent to the length of the tone-puff window) before the tone was subtracted from the averaged trace. Trial-averaged recordings were sorted by the average fluorescence during the tone-puff window for the self-sorted heat maps. To compare common responsive cells between two trial blocks, sorting was maintained between relevant heat maps in some figures.

## Calcium event detection

Onsets of calcium events can be detected using Fourier analysis, where event onset coincides with increasing low-frequency power. First, the spectrogram from each trace was calculated (MATLAB chronux, mtspecgramc with tapers = [2 3] and window = [1 0.05]), and the power below 2 Hz was averaged. To detect any significant increases in power, the change in the power at each time point ($power_{diff}$) was calculated, and then the outliers (three median absolute deviations away from the median power) in $power_{diff}$ (MATLAB function isoutlier) were identified (*Shemesh et al., 2020*).

For outliers that occurred at consecutive time points, the first outlier with positive $power_{diff}$ represented the first point of a calcium event onset. Any identified calcium event with amplitude (the signal difference between the peak and the event onset) less than seven standard deviations of the trace in the 10 s time window prior to calcium event onset was excluded. Event detection was manually inspected in about 75 traces from multiple mice to ensure validity. Baseline calcium event rate was calculated using the non-stimulus periods (defined in the section 'Spontaneous eye closure occurrence calculation').

## Determination and analysis of responsive cells

Each calcium trace was binarized such that the durations of all calcium event rising phases were assigned ones and rest of the trace was assigned zeros. We calculated the basal spontaneous calcium event amplitude distribution for each cell by shuffling the non-stimulus periods for each imaging session 1000 times. Specifically, for each shuffle, 20 non-overlapping segments of 1 s (20 frames) were randomly selected from the non-stimulus periods. The number of ones in each segment was then averaged across all 20 segments to calculate the basal calcium event amplitude for that shuffle. This procedure was repeated 1000 times to generate the baseline calcium event amplitude distribution for each cell. To determine if a cell is responsive to the CS, we calculated a cell's CS-relevant calcium event amplitude within the 1 s windows following the 20 tone onsets by averaging the number of ones in those 20 windows. If a cell's CS-relevant calcium event amplitude was greater than the 95th percentile of the baseline distribution for that cell, that cell was identified as a 'responsive cell.' 1 s was chosen as the response window to allow for the slow kinetics of calcium signaling (*Chen et al., 2013*).

For analysis of common cells between trial blocks of the late training session, 40 consecutive trials of the imaging session were split into two blocks (trials 1–20 and trials 21–40). For the mice that recieved 80 trials per day (n=3 mice), trials 21-60 of the late training session were analyzed. For mice that recieved 60 trials per day (n=6 mice), trials 1-40 of the late training session were analyzed. These chosen trials correspond to the same trial numbers that were used from the last training/extinction imaging day for each individual mouse. Common percentage between two sets of responsive cells (S1 and S2) was calculated as a percentage of total responsive cells:

$$Common\,cell\% = \frac{|S1 \cap S2|}{|S1 \cup S2|} \times 100$$

$$= \frac{C_s}{(RC_{s1} + RC_{s2} - C_s)} \times 100$$

where $C_s$ is shared (common) cells, $RC_{s1}$ is responsive cells from set 1, and $RC_{s2}$ is responsive cells from set 2. This calculation is similar to Jaccard index, which is used to assess the similarity between two finite sample sets.

## Spatial cell identity bootstrapping

Bootstraps of shuffled cell identity distributions were calculated for comparison against the observed distribution of cell identities. A 100 μm radius (76 pixels at 1.312 μm/pixel) was drawn around each cell. The number of segmented cells that existed within that spatial distribution was calculated and a percentage of either CO or EX cells was determined from the cells within that radius. For bootstrapping, the same number of CO or EX cells that was segmented for each session was randomly selected from the total population and the same calculation within a 100 μm radius was calculated. The measured percentages were then compared to the bootstrapped values for statistical confidence.

## Co-occurrence network creation

Individual trial co-occurrence matrices were created for each pair of cells across every trial. Trials 1–20 (first trial block) and trials 21–40 (second trial block) were used from the late training session. For each trial, cells that exhibited a calcium event in the 1 s following tone onset were assigned a 1, and all other cells were assigned a 0. This analysis results in a binary vector of 0s and 1s of length N, where N is the number of cells recorded in the population, for each trial. The outer product of this vector was taken with itself to yield an N × N co-occurrence matrix. This matrix is 1 if both the ith and jth cells exhibited a calcium event in the 1 s after tone onset, and 0 otherwise.

Once a co-occurrence matrix was generated for each trial, matrices could be combined for further analyses by summing certain trials of interest. For this analysis, co-occurrence matrices were summed across either the last training session, the extinction session, the first trial block, or the second trial block of the late training session. Once a trial combination matrix was created, spectral biclustering was performed for a 3 × 3 cluster pattern using the Python machine learning package scikit-learn (*Kluger et al., 2003*; *Pedregosa, 2011*). Spectral biclustering is an unsupervised clustering method initially developed for clustering microarray gene expression data (*Kluger et al., 2003*). It seeks to find checkerboard patterns within data matrices by simultaneously clustering two different features of interest (in this case, cell identities that are co-active) using spectral clustering, which determines a submatrix of the original data matrix with similar properties.

## Network map generation

Anatomical spatial information was combined with the co-occurrence matrix to generate network maps using the Python library NetworkX. The centroid of each ROI was used as the position of the corresponding node, which represent the cells of the imaging session. Since the co-occurrence matrix is symmetric, the lower triangular matrix was used to generate the edges of the network. Co-activity between two cells (any value greater than zero between those two cells in the co-occurrence matrix) was represented as an edge between the corresponding nodes. For example, the ith cell and jth cell would be connected by an edge if Ai,j is non-zero, where A is the N × N co-occurrence matrix. To analyze the most robust ensembles, in the last training and extinction session maps, edges were plotted between all cells that were deemed 'responsive' in either the last training session or the extinction session (CO and EX cells). Similarly, in the late training session maps, edges were plotted between all cells that were deemed 'responsive' in either of the trial blocks of the late training session.

## Quantification of network properties

The percentage of edges in each individual network is calculated as

$$\frac{E_{i1}}{(E_{i1} + E_{i2}) - E_s} \times 100$$

where $E_{i1}$ is the number of edges in the individual network of interest, $E_{i2}$ is the number of edges in the other network, and $E_s$ is the number of shared edges between the two networks.

The percentage of shared edges is calculated as

$$\frac{(E_{i1} + E_{i2}) - E_t}{E_t} \times 100$$

where $E_t$ is the total edges in the entire network.

Python package NetworkX was used to calculate network density, which is defined as

$$\frac{2m}{n(n-1)}$$

and degree is defined as

$$\frac{2m}{n}$$

where m is the number of edges, and n is the number of nodes.

## Acknowledgements

We thank the members of Han lab for technical support.

## Additional information

### Funding

| Funder | Grant reference number | Author |
|---|---|---|
| National Science Foundation | CBET-1848029 | Xue Han<br>Bobak Nazer |
| National Institutes of Health | 1R01MH122971-01A1 | Xue Han |
| National Institutes of Health | 1F31MH123008-01A1 | Rebecca A Mount |
| National Academy of Engineering | | Xue Han |
| National Science Foundation | DGE-1247312 | Kyle R Hansen |
| National Institutes of Health | F31 NS 105420 | Kyle R Hansen |
| National Institutes of Health | 1R21MH109941-01 | Xue Han |
| National Science Foundation | CCF-1955981 | Xue Han<br>Bobak Nazer |

The funders had no role in study design, data collection and interpretation, or the decision to submit the work for publication.

### Author contributions

Rebecca A Mount, Conducted the animal experiments, helped with data analysis, wrote the manuscript; Sudiksha Sridhar, Kyle R Hansen, Performed data analysis, wrote the manuscript; Ali I Mohammed, Conducted the animal experiments; Moona Abdulkerim, Robb Kessel, Helped with the animal experiment; Bobak Nazer, Helped with data analysis, edited the manuscript; Howard J Gritton, Conducted the animal experiment, wrote the manuscript.; Xue Han, Supervised the study, wrote the manuscript

### Author ORCIDs

Rebecca A Mount (iD) https://orcid.org/0000-0002-8962-1641
Sudiksha Sridhar (iD) https://orcid.org/0000-0003-1749-4302
Kyle R Hansen (iD) http://orcid.org/0000-0003-2782-7289
Howard J Gritton (iD) https://orcid.org/0000-0003-3194-3258
Xue Han (iD) https://orcid.org/0000-0003-3896-4609

### Ethics

Animal experimentation: All animal procedures were approved by the Boston University Institutional Animal Care and Use Committee (protocol #201800680), and all experiments were performed in accordance with the relevant guidelines and regulations.

### Decision letter and Author response

Decision letter https://doi.org/10.7554/eLife.56491.sa1
Author response https://doi.org/10.7554/eLife.56491.sa2

## Additional files

### Supplementary files

• Supplementary file 1. Marker used in all figures and cell numbers per session for each mouse.

• Supplementary file 2. Fisher's exact test for percentage of responsive cells during first session vs. last training session. n = 7 mice.

• Supplementary file 3. Fisher's exact test for percentage of responsive cells during first session vs. extinction session. n = 7 mice.

• Supplementary file 4. Fisher's exact test for percentage of responsive cells during last training session vs. extinction session. n = 9 mice.

• Supplementary file 5. Fisher's exact test for percentage of common responsive cells during both trial blocks of the late training session vs. last training and extinction sessions. n = 9 mice.

• Supplementary file 6. Fisher's exact test for percentage of shared edges during both trial blocks of the late training session vs. last training and extinction sessions. n = 9 mice.

• Transparent reporting form

### Data availability

Custom software is available at https://github.com/HanLabBU/Distinct-populations-hippocampus (copy archived at https://archive.softwareheritage.org/swh:1:rev:960e4b4d92e42697649b9b9a684ecf9c4cbb79f6).

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
