## [Decision Letter]

**Acceptance summary:**

The hippocampus is important for associative learning and memory underlying adaptive behavior. These memories can be altered by new experiences, but it is not clear if or how hippocampal neuronal ensembles dynamically encode these changes. This paper addresses this question using in-vivo calcium imaging to record the activity of hippocampal neurons during recall and subsequent extinction of an associative behavioral memory. Strikingly, the transition from stable memory expression to extinction engages a new population of neurons and reorganizes functional connectivity within neuronal assemblies in the hippocampus.

**Decision letter after peer review:**

Thank you for submitting your article "Distinct Neuronal Populations Contribute to Trace Conditioning and Extinction Learning in the Hippocampal CA1" for consideration by *eLife*. Your article has been reviewed by 3 peer reviewers, including Joshua Johansen as the Reviewing Editor and Reviewer #1, and the evaluation has been overseen by Laura Colgin as the Senior Editor. The following individual involved in review of your submission has agreed to reveal their identity: Kaori Takehara-Nishiuchi (Reviewer #2).

The reviewers have discussed the reviews with one another and the Reviewing Editor has drafted this decision to help you prepare a revised submission.

Summary:

Hansen et al. describe experiments designed to characterize how neuronal responses in the dorsal hippocampus are modulated by trace conditioning and subsequent extinction using single cell calcium imaging in mice. The authors build on previous findings that both of these behavioral tasks depend on the integrity of the dorsal hippocampus. They report that distinct CA1 cells and ensembles are recruited during conditioning and extinction, which is consistent with expectations from previous literature (eg, Lacagnina et al., 2019, Nat Neuro; Tronson et al. J Neurosci 2009; Mehrab et al. *eLife* 2014; Zhang et al. PLOS 2019). However, this is the first instance where distinct learning and extinction cells have been characterized and where identification of different neuronal ensembles have been measured in concert with learning in CA1 during trace eyeblink conditioning. This represents an advance over these previous studies.

While the results are potentially interesting, critically important controls are missing, making it difficult to draw firm conclusions from the results about whether distinct populations and networks of cells are present during eyeblink training and extinction. Furthermore, there are a number of experimental and analytical issues which need to be addressed. We therefore would like to invite the authors to respond to these concerns before a decision can be made on whether this work could be published in *eLife*.

Essential revisions:

1. The authors claim that the CS excited two largely disjoint groups of neurons during 40 CS-US paired trials (trace conditioning) and subsequent 40 CS-only trials (extinction). However, it is possible these changes are simply dependent on time (or non-associative effects), and not learning per se. To further strengthen this argument, they must demonstrate that comparable groups of neurons respond to the CS if CS-US contingency was kept the same throughout the entire session. One possibility could be that the authors use the data collected on the day before last training/extinction and the final day and examine how CS-evoked activity differed between the first 40 CS-US pairings and the last 40 CS-US pairings. If their argument is correct, the percentage of neurons with similar CS-evoked responses (Figure 2Dii) and the co-activity patterns (Figure 4E) should be comparable between the first and last 40 CS-US trials. And, these measures in the session with the contingency change should be greater than those in the control session without the contingency change. Alternatively, the authors could break up the last training session into two epochs, and run the exact same single cell and co-occurrence/network map analysis comparing these two epochs. If learning is the driving factor behind the emergence of distinct ensembles over time, the percentage of overlapping cells and percentage of cells with shared edges should be larger when comparing epochs that both occurred during the last session vs comparing either of these to extinction. A final possibility would be to run another control group in which the final extinction session is replaced with further overtraining to examine whether the extinction induced changes are produced by extinction or some other non-associative process. In this case cell and ensemble (co-active cells) could be examined across the last session of training and the subsequent overtraining. The prediction would be that there would not be as much change in the cell/ensemble representation with overtraining compared with extinction. Something like this is required to make any claims regarding differences in cell populations during training and extinction.

2. On a similar note, the comparison between correct and incorrect trials needs controls. We would recommend that the authors run the same analysis (Figure 5) after shuffling the trial labels (i.e., correct or incorrect trials) assigned to each co-activity pattern. This analysis allows for testing whether the percentage of the shared edge (Figure 5c) between correct and incorrect trials is significantly lower than the chance level.

3. The method of cell registration across sessions is poorly described. As the quality of cross-registration is crucial for some aspects of this study (within session cross registration is less of a worry), evidence for correct registration is key. The stated method of registering cells within 50 pixels (~50um) of one another and having at least 50% of pixel overlap seems like it has the potential for many false positives, particularly with 1-photon imaging where the authors are effectively sampling cells across tens of μm in the z-dimension. How was registration validated? Also, it is unclear as to whether registration was performed across the last session and extinction session.

4. There is considerable variability among mice for the number of cells examined, yet each mouse's contribution is not weighted in the analysis to reflect this. Can the authors show that the results of their analyses are not sensitive to differing numbers of cells across animals? Otherwise, can they provide alternative analyses that are not subject to this influence (e.g., pool all neurons from all mice for a session type and run a fisher's exact test (or equivalent) to test for significance when assessing changes in the proportion of neural responses across session types (e.g., first vs last training session)). Alternatively using subsampling procedures using numbers from the mouse with the least number of cells could address this issue.

5. Similarly, there is considerable variability among mice for behavioral performance (Figure 2B). Is there a correlation between overall task performance and neural responses? All individual data points should be color coded for mouse identity so the reader can track how behavioral performance corresponds to neural data.

6. Although the absolute value of CR% varies depending on the criteria used to detect CRs, CR% of ~40% in the first session seems very high. Please include the frequency of spontaneous eyeblinks in Figure 2B for comparison.

7. For the data shown in Figure 2, a more detailed neuronal analysis of the response properties of CA1 cells is necessary. One important point is that extinction is not a static process, while, in theory, the last training session occurring before it is. One might imagine that during early extinction the network would be more similar to the last training session and that over the course of extinction the extinction defined cells would emerge. As it stands, the extinction session is lumped into one category. The authors should analyze this in more detail by determining the change in response properties over the course of extinction and comparing different portions of the extinction dataset to the last training data. Furthermore, it says that the data in Figure 2C comes from one example animal. The authors should show a heat plot for all cells from all animals to give a more comprehensive view. In addition, in Figure 2C most of the responsive cells have a high activity level prior to stimulus onset. This is confusing as it says in the methods that the cells were classified as having a larger response during the stimulus period compared with the pre-stimulus period. Furthermore, it is unclear why only cells with higher baseline/pre-stimulus activity would be the 'responsive' cells. Additionally, it would be helpful to show a population averaged peri-event time histogram of the responses in the different trial-periods shown in Figure 2C to give a better idea of the dynamics. Finally, for Figure 2D, the authors should say what the data is normalized to.

8. Is there a reason why the trial sample size is not matched across comparisons? For example, incorrect trials constitute, on average, only ~1/4 the amount of correct trials. Given the large variability in trial-to-trial neural activity, the number of edges identified via co-occurence analysis would likely increase with trial number, which could be driving the false impression of greater connectivity (albeit short of significance) during correct vs incorrect trials. Since for most animals the number of correct trials will be >2x the number of incorrect trials, the authors could also take subsets of correct trials and compare these against each other to test for consistency across correct trials.

9. There are several instances of 1-tailed tests throughout the manuscript (e.g., when comparing the proportion of tone-trace active cells during training vs extinction). A rationale for these 1-tailed tests should be provided or, ideally, 2-tailed tests should be used.

10. The authors should address why the specific value of 2% deviation from average eye size was chosen as the threshold for a conditioned response? Is this an arbitrary value? Empirically based?

11. Regarding the co-occurence network map, the authors should address whether the threshold for edge creation between two nodes just a single trial where the 2 nodes were coactive. If they increase this threshold so only more robust ensembles are counted, does the proportion of shared edges change appreciably?

12. The method of fluorescence trace normalization in which they fit a normal distribution to the lowest 50th percentile of the data, used the mean of this distribution as the baseline , subtracted this baseline from each trace and then scaled data by 5% of the maximum range of the full calcium trace seems somewhat convoluted and suggests that they cannot isolate single cells/ROIs. This should be explained and/or corrected.

13. The method of classifying a cell as responsive during the tone-puff window (described in lines 526-538) is not ideal. The classification threshold of a larger calcium response during the stimulus compared with the baseline period of 0.15 (several 5% or one 30% increase) seems somewhat arbitrary. It would be better to use a statistical comparison.

References:

Lacagnina et al. Distinct hippocampal engrams control extinction and relapse of fear memory. Nature Neuroscience 22, 2019

Mehrab et al. CA1 cell activity sequences emerge after reorganization of network correlation structure during associative learning. *eLife* 2014

Tronson et al. Segregated Populations of Hippocampal Principal CA1 Neurons Mediating Conditioning and Extinction of Contextual Fear. J. Neurosci. 29, 2009

Zhang et al. Dynamics of a hippocampal neuronal ensemble encoding trace fear memory revealed by in vivo Ca^2+^ imaging. PLOS One 2019

[Editors' note: further revisions were suggested prior to acceptance, as described below.]

Thank you for submitting your article "Distinct Neuronal Populations Contribute to Trace Conditioning and Extinction Learning in the Hippocampal CA1" for consideration by *eLife*. Your article has been reviewed by 3 peer reviewers, including Joshua Johansen as the Reviewing Editor and Reviewer #1, and the evaluation has been overseen by Laura Colgin as the Senior Editor. The following individual involved in review of your submission has agreed to reveal their identity: Kaori Takehara-Nishiuchi (Reviewer #2).

The reviewers have discussed the reviews with one another and the Reviewing Editor has drafted this decision to help you prepare a revised submission.

Summary:

The authors have added new data and analyses which provide stronger support for the claims made in the paper. However, there are some remaining concerns which should be addressed before a final decision can be made.

Essential revisions:

1. The greatest concern with the initial submission was, that the authors show that distinct populations are engaged during late training vs extinction periods, but it is possible these changes are simply dependent on time (or non-associative effects, and not learning per se). The reviewers suggested two potential solutions to this: "One possibility could be that the authors use the data collected on the day before last training/extinction and the final day and examine how CS-evoked activity differed between the first 40 CS-US pairings and the last 40 CS-US pairings… Alternatively, the authors could break up the last training session into two epochs, and run the exact same co-occurrence/network map analysis comparing these two epochs." This additional analysis of splitting up the prior days session was run. However, the analysis of this is unclear and it could still benefit from modification to clarify and/or optimally address the concern it was in response to. Specifically, in Figure 2, the authors compare neural activity in the 'First Half' and 'Second Half' of Late Training, but it is unclear what trials these represent. Relatedly, when comparing Last Training to Extinction it is not clear what trials were used. This is confusing because they use either 60 or 80 trials on the Late Training day, 20 or 40 trials for the Last Training session and 40 Extinction trials. The ideal analysis would be to use 60 trials for late training, 20 last training trials and 40 extinction trials so that the late training (20 early, 40 late) comparison could be matched to the last day (20 last training, 40 extinction). The authors need to explain how this was done and clearly state this in the text/figure legend. If possible, they should also try to avoid gaps in trials in, for example, the Late Training or Last Training session to match trial numbers as this could present a confound.

2. Related to point 1, the authors state that 3 mice had an approx. 20 minute wait between the end of the CS-US session and the CS-only extinction session, where the animals remained headfixed. It introduces an additional 20-minute gap that is not present in the late session control data. Should these animals be excluded from analysis? Alternatively, they could analyze the data both ways and if there aren't major differences then combine them.

3. The authors have modified their criteria for task-responsive cells, but it still leaves much to be desired. The criteria of being active on 10%+ of trials was selected as a threshold because it is 3x the average background firing rate of ALL cells. However, cells with a higher baseline firing rate would be more likely to be labeled as responsive, just by virtue of their increased firing probability (and not due to any specific responsivity to the CS or trace). Why not tailor the cutoff on a cell by cell basis? Eg, run statistical analysis comparing CS/trace activity to background activity on a cell-by-cell basis? Or compare a cell's CS/trace activity to a shuffled distribution of that cell's background activity?

4. In the initial version of the manuscript there was a significant behavioral extinction detected, but in the current version there is no significant behavioral extinction (Figure 2B in previous and current versions and explained in lines 285-293). This change should be explained to the reviewers. Also, this is a bit of a problem for the interpretation of their data which relies on trial summed neural responses. They may be able to resolve this by statistically comparing the last training session CR to the later extinction session (similar to the first half to last half extinction analysis reported in lines 211-217). Furthermore, they could/should also show the trial-by-trial CRs during the entire extinction session (single trial or 2/3 trial bins). This may provide support for their contention that "…learning is dynamic during the extinction session" and give the reader more insight into the behavioral changes occurring during extinction.

[Editors' note: further revisions were suggested prior to acceptance, as described below.]

Thank you for submitting your article "Distinct Neuronal Populations Contribute to Trace Conditioning and Extinction Learning in the Hippocampal CA1" for consideration by *eLife*. Your article has been reviewed by 2 peer reviewers, including Joshua Johansen as the Reviewing Editor and Reviewer #1, and the evaluation has been overseen by Laura Colgin as the Senior Editor.

Summary:

The authors have addressed most of the reviewer concerns, but there is one outstanding reviewer comment which we advise you to consider and incorporate into the final version of your manuscript (see below).

Essential revisions:

We thank the authors for the clarification regarding point number 1 of the essential revisions. The only remaining suggestion is that the authors incorporate the gap between the late session trial blocks used for the control Jaccard index comparison or discuss this issue in the conclusion of the paper. That is, instead of comparing trials 1-20 and 21-40, compare trials 1-20 and 61-80 (or 41-60 in animals that did not undergo 80 trials). The idea is for this control analysis to better reflect the gap between last and extinction session blocks (which appears to be either 40 trials, or presumably ~ 20 mins + 20 trials for the animals that were subject to the technical memory limitation), and thus more precisely control for any potential changes due to elapsed time or task engagement.

*Reviewer #1:*

The authors have addressed my previous concerns. I support publication.

*Reviewer #3:*

The authors have addressed the majority of previous concerns. The only remaining concern is the analysis of the late session, where the authors should incorporate the gap between the late session trial blocks used for the control Jaccard index comparison. That is, instead of comparing trials 1-20 and 21-40, compare trials 1-20 and 61-80 (or 41-60 in animals that did not undergo 80 trials). The idea is for this control analysis to better reflect the gap between last and extinction session blocks (which appears to be either 40 trials, or presumably ~ 20 mins + 20 trials for the animals that were subject to the technical memory limitation), and thus more precisely control for any potential changes due to elapsed time or task engagement.

---

## [Author Response]

Essential revisions:1. The authors claim that the CS excited two largely disjoint groups of neurons during 40 CS-US paired trials (trace conditioning) and subsequent 40 CS-only trials (extinction). However, it is possible these changes are simply dependent on time (or non-associative effects), and not learning per se. To further strengthen this argument, they must demonstrate that comparable groups of neurons respond to the CS if CS-US contingency was kept the same throughout the entire session. One possibility could be that the authors use the data collected on the day before last training/extinction and the final day and examine how CS-evoked activity differed between the first 40 CS-US pairings and the last 40 CS-US pairings. If their argument is correct, the percentage of neurons with similar CS-evoked responses (Figure 2Dii) and the co-activity patterns (Figure 4E) should be comparable between the first and last 40 CS-US trials. And, these measures in the session with the contingency change should be greater than those in the control session without the contingency change. Alternatively, the authors could break up the last training session into two epochs, and run the exact same single cell and co-occurrence/network map analysis comparing these two epochs. If learning is the driving factor behind the emergence of distinct ensembles over time, the percentage of overlapping cells and percentage of cells with shared edges should be larger when comparing epochs that both occurred during the last session vs comparing either of these to extinction. A final possibility would be to run another control group in which the final extinction session is replaced with further overtraining to examine whether the extinction induced changes are produced by extinction or some other non-associative process. In this case cell and ensemble (co-active cells) could be examined across the last session of training and the subsequent overtraining. The prediction would be that there would not be as much change in the cell/ensemble representation with overtraining compared with extinction. Something like this is required to make any claims regarding differences in cell populations during training and extinction.

We thank the reviewers for pointing out this important control. We agree with the recommendation of the reviewer, and as suggested, we analyzed the data collected on the day before the last training/extinction session (referred to as “late training day” here and in the manuscript) in the same way that we analyzed the last training session and extinction session. Briefly, we split the late training day into two blocks of 20 trials (“first half” and “second half”). We identified responsive cells in the same way that we identified populations for the last training and extinction sessions, and determined the percentage of cells that were common between the first half and second half responsive populations. Common cell percentage is significantly higher between the two halves of the late training session than between the last training session and extinction session, confirming that there are distinct populations that underlie conditioning and extinction. These results are summarized in Figure 2 and the Results section entitled “Extinction learning rapidly recruits new CA1 neurons.”

Additionally, we generated co-activity matrices for the two blocks of the late training session, and compared the edges shared between the networks built from these two matrices. We found that shared edges are significantly higher between the two halves of the late training session than between the last training session and extinction session, confirming that there are distinct sub-networks that underlie conditioning and extinction. These results are summarized in Figure 4, Figure 4—figure supplement 2, and the Results section entitled “Co-occurrence analysis reveals differential connectivity between sub-populations of neurons during trace conditioning versus extinction learning.”

2. On a similar note, the comparison between correct and incorrect trials needs controls. We would recommend that the authors run the same analysis (Figure 5) after shuffling the trial labels (i.e., correct or incorrect trials) assigned to each co-activity pattern. This analysis allows for testing whether the percentage of the shared edge (Figure 5c) between correct and incorrect trials is significantly lower than the chance level.

Unfortunately, calcium event rates are sparse, and the animals’ high percentages of correct responses means that all animals have 9 or fewer incorrect trials (one animal has only 3). As such, we do not have sufficient data to compare these conditions statistically. Thus, we have removed analysis between correct and incorrect trials, and Figure 5, from the revised manuscript. We now focus on differences between populations involved in the conditioning and extinction sessions.

3. The method of cell registration across sessions is poorly described. As the quality of cross-registration is crucial for some aspects of this study (within session cross registration is less of a worry), evidence for correct registration is key. The stated method of registering cells within 50 pixels (~50um) of one another and having at least 50% of pixel overlap seems like it has the potential for many false positives, particularly with 1-photon imaging where the authors are effectively sampling cells across tens of μm in the z-dimension. How was registration validated? Also, it is unclear as to whether registration was performed across the last session and extinction session.

Co-registration was necessary for three of the nine mice (mice #1-3); these mice received 40 trials as the last training session and 40 trials as the extinction session, for a total of 80 trials. To ensure a large field of view and high speed imaging, images collected from the sCMOS camera were stored in computer random access memory (RAM) in real time, and then transferred to hard drive storage after image collection. Due to computer memory limitations, imaging needed to be paused between the 40 conditioning trials and 40 extinction trials in order to transfer the recorded video from computer memory to hard drive storage. This transfer takes ~20 minutes. During this data transfer time, neither the mouse nor the microscope were touched or moved. Once the data transfer was complete, we immediately resumed imaging to begin the extinction session. During data processing, these two videos were motion corrected and segmented for neuronal ROIs separately. Co-registration was then performed automatically as described, and validated by visual inspection of visible static landmarks (blood vessel patterns, specific cells with distinctive shapes, etc.). We are confident that co-registration did not introduce significant errors because the mice and microscope did not move, most neurons remained in both videos, and results are similar to mice with continuous recordings. Nonetheless, to avoid co-registration in the other six mice, we subsequently performed imaging for a total of 60 trials per imaging session (20 last session trials and 40 extinction session trials) to avoid the computer memory limitation. This video was continuous over all 60 trials, and thus no co-registration was needed between the last training and extinction sessions (n=6 mice, mice #4-9). We have now added these details to the Methods section “Neuronal Trace Extraction.”

4. There is considerable variability among mice for the number of cells examined, yet each mouse's contribution is not weighted in the analysis to reflect this. Can the authors show that the results of their analyses are not sensitive to differing numbers of cells across animals? Otherwise, can they provide alternative analyses that are not subject to this influence (e.g., pool all neurons from all mice for a session type and run a fisher's exact test (or equivalent) to test for significance when assessing changes in the proportion of neural responses across session types (e.g., first vs last training session)). Alternatively using subsampling procedures using numbers from the mouse with the least number of cells could address this issue.

We thank the reviewer for suggesting a Fisher’s exact test for analyzing the proportion of neurons that respond across sessions, which is a more appropriate statistical test. We have included this statistical test in Figures 2Ei, 2Eii, and 4Eii. The Fisher’s exact test contingency tables for these figures are included as supplementary files 2-6.

5. Similarly, there is considerable variability among mice for behavioral performance (Figure 2B). Is there a correlation between overall task performance and neural responses? All individual data points should be color coded for mouse identity so the reader can track how behavioral performance corresponds to neural data.

We have edited all bar plots to include unique data markers for each mouse as the reviewer suggested. The marker assignments are summarized in supplementary file 1. Unfortunately, because of the small differences in performance for the majority of mice, we were not able to test correlation strength between neural responses and behavioral performance.

6. Although the absolute value of CR% varies depending on the criteria used to detect CRs, CR% of ~40% in the first session seems very high. Please include the frequency of spontaneous eyeblinks in Figure 2B for comparison.

We have added analysis of spontaneous blink occurrence to the manuscript as the reviewer suggested. A sliding 600ms window of the non-stimulus periods was analyzed for a blink in the same way that each tone-puff window was analyzed. Spontaneous blink occurrence is consistent at about 20% during the first, last, and extinction sessions. We believe that the 40% response rate during the first session reflects learning within the training session. We also noticed that 2 of the 9 mice exhibited very rapid learning, and correctly responded to more than 75% of trials in the first session. As such, they were excluded from analyses that compare the first session and the last training session. Spontaneous blink occurrence is summarized in the new Figure 2—figure supplement 1 and described in the new Methods section entitled “Spontaneous Eye Closure Occurrence Calculation.” Exclusion of the 2 rapid learners is described in the corresponding Results section (“Conditioned responding increases across trace conditioning sessions in a classical eye blink task and decreases during extinction learning”), and also detailed in the Methods section entitled “Eye-Blink Segmentation and Analysis.”

7. For the data shown in Figure 2, a more detailed neuronal analysis of the response properties of CA1 cells is necessary. One important point is that extinction is not a static process, while, in theory, the last training session occurring before it is. One might imagine that during early extinction the network would be more similar to the last training session and that over the course of extinction the extinction defined cells would emerge. As it stands, the extinction session is lumped into one category. The authors should analyze this in more detail by determining the change in response properties over the course of extinction and comparing different portions of the extinction dataset to the last training data.

We agree with the recommendation of the reviewer, and we have included new heat maps (from the same animal shown in Figure 2) that show the evolution of the population response from last training session to extinction session (new Figure 2—figure supplement 5). These heat maps were generated with 10 trials each. All heat maps in Figure 2—figure supplement 5A are sorted by the average fluorescence during the tone-puff window of the first 10 trials of the last training session. The responsive population is mostly maintained during the last training session, and many of these cells continue to respond during the first 10 trials of extinction. However, over the course of extinction learning, some cells stop responding. The emergence of the extinction population can be seen in Figure 2—figure supplement 5B, as all the heat maps are sorted by the average fluorescence during the tone-puff window of the last 10 trials (trials 31-40) of the extinction session. Based on these results, we have revised our analysis to use the last 20 trials of the extinction session for responsive cell analyses, to most accurately capture the population that has emerged to respond during extinction only.

Furthermore, it says that the data in Figure 2C comes from one example animal. The authors should show a heat plot for all cells from all animals to give a more comprehensive view.

We have included heat maps that include all cells from all animals for the first, last, and extinction sessions, in new Figure 2—figure supplement 3.

In addition, in Figure 2C most of the responsive cells have a high activity level prior to stimulus onset. This is confusing as it says in the methods that the cells were classified as having a larger response during the stimulus period compared with the pre-stimulus period. Furthermore, it is unclear why only cells with higher baseline/pre-stimulus activity would be the 'responsive' cells. Additionally, it would be helpful to show a population averaged peri-event time histogram of the responses in the different trial-periods shown in Figure 2C to give a better idea of the dynamics. Finally, for Figure 2D, the authors should say what the data is normalized to.

The original Figure 2C was sorted by average fluorescence during the tone-puff window to allow for easy visualization of the population responses. Thus, cells that have a naturally high baseline fluorescence (simply due to cell-to-cell variability) will be sorted to the bottom of those plots, even if they are not deemed a “responsive cell.” We agree that cells with high baseline fluorescence is distracting and misleading, so for visualization purposes only, all heat maps shown in the main figures and figure supplements are now “baseline subtracted.” Each cell’s average fluorescence value from the 600ms prior to tone onset is subtracted from that cell’s trace (from 1 second before to 3 seconds after tone onset). Discussion of this change is included in the Methods section entitled “Fluorescence Trace Normalization.” New Figure 2Ei (originally Figure 2D) is normalized to cell number per mouse. The data is shown as a percentage of total cells from each mouse in the relevant session.

8. Is there a reason why the trial sample size is not matched across comparisons? For example, incorrect trials constitute, on average, only ~1/4 the amount of correct trials. Given the large variability in trial-to-trial neural activity, the number of edges identified via co-occurence analysis would likely increase with trial number, which could be driving the false impression of greater connectivity (albeit short of significance) during correct vs incorrect trials. Since for most animals the number of correct trials will be >2x the number of incorrect trials, the authors could also take subsets of correct trials and compare these against each other to test for consistency across correct trials.

The figure comparing correct and incorrect trials (formerly Figure 5) has been removed (see response to comment 2 above).

9. There are several instances of 1-tailed tests throughout the manuscript (e.g., when comparing the proportion of tone-trace active cells during training vs extinction). A rationale for these 1-tailed tests should be provided or, ideally, 2-tailed tests should be used.

We thank the reviewer for pointing this out. As suggested, we have changed all relevant statistical tests to 2-tailed tests.

10. The authors should address why the specific value of 2% deviation from average eye size was chosen as the threshold for a conditioned response? Is this an arbitrary value? Empirically based?

The 2% deviation threshold was selected after plotting the eye blink traces. Blinks (conditioned responses) were both manually identified and automatically identified by threshold crossing, and manually compared. 2% was the value determined to match what would have been manually labelled as blinks by an independent observer. Discussion of the selection of this threshold has been added to the Methods section “Eye-Blink Segmentation and Analysis.”

11. Regarding the co-occurence network map, the authors should address whether the threshold for edge creation between two nodes just a single trial where the 2 nodes were coactive. If they increase this threshold so only more robust ensembles are counted, does the proportion of shared edges change appreciably?

We have changed the network maps shown in Figure 4 to include more robust ensembles by plotting edges only between cells that were identified as responsive cells during either the last training session or the extinction session. Similarly, the newly-included network maps for the late training session (Figure 4—figure supplement 2) also show more robust ensembles, as edges were only plotted between cells that were deemed responsive during the first half or second half of the late training session. All of these cells were active on at least 10% of trials. The number of common edges between the last training and extinction session maps is significantly lower than the number of common edges between the first half and second half late training session maps, as summarized in Figure 4Eii.

12. The method of fluorescence trace normalization in which they fit a normal distribution to the lowest 50th percentile of the data, used the mean of this distribution as the baseline, subtracted this baseline from each trace and then scaled data by 5% of the maximum range of the full calcium trace seems somewhat convoluted and suggests that they cannot isolate single cells/ROIs. This should be explained and/or corrected.

We apologize for any confusion our original description created. Normalization analysis is performed on each identified cell after ROI segmentation. It is unrelated to single cell/ROI segmentation, which is performed using the maximum-minus-minimum projection image of the whole video. We have expanded discussion of normalization in the Methods section entitled “Fluorescence Trace Normalization” and we have added Figure 1—figure supplement 1 to further explain this technique. Briefly, the mean of the lowest 50th percentile of the calcium trace is used as an estimate of baseline fluorescence level, which excludes data points corresponding to calcium spikes that have fluorescence levels in the upper 50th percentile. The lowest 50th percentile of the calcium trace for one neuron follows a parametric Gaussian distribution (validated by quantile-quantile plots of data vs. a normal distribution, Figure 1—figure supplement 1A), and thus allows us to estimate true baseline of the fluorescence in the absence of calcium spikes. Since raw GCaMP6 fluorescence varies among simultaneously recorded neurons, we used this baseline fluorescence level to normalize the fluorescence of each neuron to 100%, so that we can average fluorescence traces across neurons in the same session to visualize population changes on the same scale.

In our original manuscript, we scaled calcium data to 5% of the maximum range of each trace in order to more easily represent statistically significant fluorescence values in Figure 2. With this scaling factor, the scale of fluorescence ranged from 0-20. However, we agree that this scaling is confusing and unnecessary. We have thus removed the scaling step from data processing and edited all normalized fluorescence scale bars to reflect this change. Normalized fluorescence is now presented as a percentage of each neuron’s maximum fluorescence.

13. The method of classifying a cell as responsive during the tone-puff window (described in lines 526-538) is not ideal. The classification threshold of a larger calcium response during the stimulus compared with the baseline period of 0.15 (several 5% or one 30% increase) seems somewhat arbitrary. It would be better to use a statistical comparison.

We agree with the reviewer, and have revised our responsive cell classification technique. Cells are now deemed to be activated on a given trial if they have 1 calcium onset event in the 1-second window following tone onset. If a cell is activated on ≥10% of trials, it is classified as a responsive cell. The 1-second window for responsivity was chosen to account for the slow GCaMP fluorescence dynamics. 10% of trials was chosen as the relevant threshold because this level of activation is equivalent to 0.1 events/sec, which is more than three times higher than the basal spontaneous event rate (0.03 events/sec during the non-stimulus periods) for all mice. We have added two new Methods sections entitled “Event Detection” and “Determination of Responsive Cells,” and we have included new Figure 2—figure supplement 2 to visualize event detection. We also included a brief description of this classification technique in the corresponding Results section (“Calcium dynamics in CA1 neurons track behavioral responses during trace conditioning”).

[Editors' note: further revisions were suggested prior to acceptance, as described below.]

Essential revisions:1. The greatest concern with the initial submission was, that the authors show that distinct populations are engaged during late training vs extinction periods, but it is possible these changes are simply dependent on time (or non-associative effects, and not learning per se). The reviewers suggested two potential solutions to this: "One possibility could be that the authors use the data collected on the day before last training/extinction and the final day and examine how CS-evoked activity differed between the first 40 CS-US pairings and the last 40 CS-US pairings… Alternatively, the authors could break up the last training session into two epochs, and run the exact same co-occurrence/network map analysis comparing these two epochs." This additional analysis of splitting up the prior days session was run. However, the analysis of this is unclear and it could still benefit from modification to clarify and/or optimally address the concern it was in response to. Specifically, in Figure 2, the authors compare neural activity in the 'First Half' and 'Second Half' of Late Training, but it is unclear what trials these represent. Relatedly, when comparing Last Training to Extinction it is not clear what trials were used. This is confusing because they use either 60 or 80 trials on the Late Training day, 20 or 40 trials for the Last Training session and 40 Extinction trials. The ideal analysis would be to use 60 trials for late training, 20 last training trials and 40 extinction trials so that the late training (20 early, 40 late) comparison could be matched to the last day (20 last training, 40 extinction). The authors need to explain how this was done and clearly state this in the text/figure legend. If possible, they should also try to avoid gaps in trials in, for example, the Late Training or Last Training session to match trial numbers as this could present a confound.

We apologize for any confusion that our original descriptions created. We have now revised Figure 1B to include the late training session. To standardize analysis across all sessions and all animals, we examined the final 20 trials of each session for the first training session, last training session, and extinction session. We have added discussion of these trial blocks to the Results section related to Figure 1, and the trial blocks are also noted in the Methods sections entitled “Eye-Blink Segmentation and Analysis.” For the first session, the final 20 trials were used to avoid potential startle responses to novel stimuli in the mice (described in the Methods section “Eye-Blink Segmentation and Analysis”). For the extinction session, the final 20 trials were used to capture the most robust extinction learning, as learning emerges during the session (as described in the Results section entitled “Conditioned responding increases across trace conditioning sessions in a classical eye blink task and decreases during extinction learning”). For the last training session, we used the final 20 trials to minimize any temporal gaps. Finally, for the late training session, two consecutive blocks of 20 trials were used for analysis (trials 1-20 and trials 21-40 of the session). We have changed all wording in the paper from “first half” and “last half” of the late training session to “first trial block” and “second trial block” to more accurately reflect the breakdown into 20 trial groupings. Also, we now clearly state this trial breakdown in the Results sections entitled “Extinction learning rapidly recruits new CA1 neurons” and “Co-occurrence analysis reveals differential connectivity between sub-populations of neurons during trace conditioning versus extinction learning.” as well as the Methods sections entitled “Determination and Analysis of Responsive Cells” and “Co-Occurrence Network Creation.”

As mentioned above, we analyzed the final 20 trials of the extinction session (extinction trials 21-40), instead of the first 20 extinction trials (extinction trials 1-20), because extinction learning occurs at different rates across animals. Thus, we felt that analyzing the final 20 extinction trials provides the most consistent measure of the effects of extinction learning on neural activity across all animals (also see response to comment 4 below). This decision was based largely on analysis of the first 20 extinction trials, which occur immediately after conditioning. We found that the common cell Jaccard index for the last training session and the first 20 trials of the extinction session (13.2%) is higher than the Jaccard index for the last training session and the final 20 trials of the extinction session (11.1%). However, both of these indices are lower than the Jaccard index for the late training session (19.9%). These analyses indicate that during the first 20 trials of the extinction session, different cells begin to be recruited in response to the tone, and this trend becomes stronger as extinction continues. Thus, we feel that analyzing the final 20 extinction trials is more appropriate in describing the neuronal effect of extinction learning, while still streamlining statistical comparisons among various trial blocks as we have performed in the manuscript.

The above analysis also indicates that gaps between analyzed trial blocks do not drastically affect our results, as the effects seen between the last training session and extinction trials are the same whether the first 20 extinction trials or final 20 extinction trials are used for comparison.

2. Related to point 1, the authors state that 3 mice had an approx. 20 minute wait between the end of the CS-US session and the CS-only extinction session, where the animals remained headfixed. It introduces an additional 20-minute gap that is not present in the late session control data. Should these animals be excluded from analysis? Alternatively, they could analyze the data both ways and if there aren't major differences then combine them.

We compared the behavioral performance of the three animals with the gap to the animals without the gap, as suggested by the reviewer. These animals’ behavioral responses closely follow the overall trend that also includes animals without the temporal gap, therefore, we grouped all animals together for analysis. We have highlighted the markers for the three animals with a temporal gap in green in Author response image 1, which is taken from Figure 2B.

3. The authors have modified their criteria for task-responsive cells, but it still leaves much to be desired. The criteria of being active on 10%+ of trials was selected as a threshold because it is 3x the average background firing rate of ALL cells. However, cells with a higher baseline firing rate would be more likely to be labeled as responsive, just by virtue of their increased firing probability (and not due to any specific responsivity to the CS or trace). Why not tailor the cutoff on a cell by cell basis? Eg, run statistical analysis comparing CS/trace activity to background activity on a cell-by-cell basis? Or compare a cell's CS/trace activity to a shuffled distribution of that cell's background activity?

We thank the reviewer for suggesting the inclusion of a more robust criterion for identifying responsive cells, and we have changed our method of identifying these cells as suggested by the reviewer. The results influenced by this new method are plotted in Figure 2E, as in the previous version of the manuscript. The details of the new criterion are described in both the Results section entitled “Calcium dynamics in CA1 neurons track behavioral responses during trace conditioning” and the Methods section entitled “Determination and Analysis of Responsive Cells.” Specifically, we first binarized each calcium trace by assigning 1s to the rising phases of all calcium events and 0s to the rest of the trace. We then randomly selected 20 1-second windows of the binarized calcium event traces from the non-stimulus periods of each imaging session (“non-stimulus periods” include the time periods that are more than 2 seconds before tone onset or more than 8 seconds after tone onset). From these randomly selected 20 windows, we calculated a basal spontaneous calcium event amplitude by averaging the number of 1s across the 20 windows. We repeated this procedure 1000 times for each trace to build a basal spontaneous calcium event amplitude distribution, which normalizes each cell by its own spontaneous event rate. To determine if a cell is responsive to the tone (CS), we calculated the calcium event amplitude within 1-second windows following the 20 tone onsets. If a cell’s CS-relevant calcium event amplitude was greater than the 95^th^ percentile of the basal distribution for that cell, that cell was identified as a “responsive cell.”

4. In the initial version of the manuscript there was a significant behavioral extinction detected, but in the current version there is no significant behavioral extinction (Figure 2B in previous and current versions and explained in lines 285-293). This change should be explained to the reviewers. Also, this is a bit of a problem for the interpretation of their data which relies on trial summed neural responses. They may be able to resolve this by statistically comparing the last training session CR to the later extinction session (similar to the first half to last half extinction analysis reported in lines 211-217). Furthermore, they could/should also show the trial-by-trial CRs during the entire extinction session (single trial or 2/3 trial bins). This may provide support for their contention that "…learning is dynamic during the extinction session" and give the reader more insight into the behavioral changes occurring during extinction.

In the initial submission of the manuscript, behavioral extinction was analyzed using 40 trials. In the previous revision and current revision, for standardization purposes as detailed in response to point 1, behavioral extinction was analyzed using only the last 20 trials of the extinction session; thus, some statistical power is lost between the two versions. Additionally, in the original version of the manuscript, the statistical test used was a one-tailed t-test. At the suggestion of the reviewers, we changed this test to two-tailed in the previous revision, which resulted a borderline p value of 0.052. Upon further evaluation of these behavioral data, however, we noted that the behavioral data was not normally distributed. As such, we have now implemented the non-parametric Wilcoxon rank-sum test, instead of the two-tailed t-test, in the current revision. Using the non-parametric version of the test, we found significantly reduced conditioning responding during the extinction session as compared to the last training session (p=0.034). We note that during extinction, most animals showed a reduction in conditioned response (CR) rate, and thus some level of extinction occurred in most animals. While two animals showed a small increase in CR rate, suggesting delayed extinction learning, we chose to include all animals in this analysis since changes in network activity may precede changes seen in behavioral performance.

To show the progression of behavioral extinction through the session as requested by the reviewer, we have included a plot of the average CR rate within a sliding window of 10 trials from all 40 trials of the extinction session, averaged across all 9 animals (Author response image 2). Error shading is shown as standard error to the mean. While behavioral performance is slightly variable, CR rate is, on average, lower during the final 20 trials of the extinction session as compared to the first 20 trials. Because of the variations between animal learning acquisition and trial-by-trial responding, it is difficult to compare performance using a small number of trials from discrete windows. Thus we kept Figure 2B as an average of 20 trials, where the window of performance from the population is the most consistent.

**Author response image 2. respfig2:** 

[Editors' note: further revisions were suggested prior to acceptance, as described below.]

Essential revisions:We thank the authors for the clarification regarding point number 1 of the essential revisions. The only remaining suggestion is that the authors incorporate the gap between the late session trial blocks used for the control Jaccard index comparison or discuss this issue in the conclusion of the paper. That is, instead of comparing trials 1-20 and 21-40, compare trials 1-20 and 61-80 (or 41-60 in animals that did not undergo 80 trials). The idea is for this control analysis to better reflect the gap between last and extinction session blocks (which appears to be either 40 trials, or presumably ~ 20 mins + 20 trials for the animals that were subject to the technical memory limitation), and thus more precisely control for any potential changes due to elapsed time or task engagement.

We agree with the reviewer that for each mouse, the trial blocks, and the gaps between those blocks, should be identical between the last/extinction sessions and late training session to “precisely control for any potential changes due to elapsed time or task engagement”. We have now addressed this by removing gaps in our trial block comparisons, and we compare the exact same trial blocks during the last training/extinction sessions and late training session. In detail, we now analyze 40 consecutive trials. On the last imaging day, we analyzed the 20 CS-US conditioning trials immediately preceding the extinction trials (“last training session”), and the first 20 CS-only extinction trials immediately after the last training session (“extinction session”). There is no gap between the last training and extinction sessions. Similarly, on the late training day, we analyzed 40 consecutive trials, using the same corresponding trial blocks for each individual mouse analyzed. We have now added these details to the Methods section and we have updated corresponding results from this new analysis. With this new analysis, we found that the differences in neuronal dynamics between the last training and extinction sessions are now more pronounced and significant. These results provide even stronger support of our claim of distinct populations of cells involved in these two learning processes. We would like to thank the reviewer for this excellent suggestion, which has further improved our manuscript.